# The locking mechanism of human TRPV6 inhibition by intracellular magnesium

Arthur Neuberger [1,7], Alexey Shalygin[2,7], Irina I. Veretenenko [3,7], Yury A. Trofimov [3,4], Thomas Gudermann [2,5], Vladimir Chubanov [2] ✉, Roman G. Efremov [3,4,6] ✉ & Alexander I. Sobolevsky [1] ✉

TRPV6 is a member of the vanilloid subfamily of transient receptor potential channels, which serves as the master regulator of $Ca^{2+}$ homeostasis. TRPV6 functions as a constitutively active $Ca^{2+}$ channel, and emerging evidence indicates that its overactivity underpins the progression of several human diseases, including cancer. Hence, there is a pressing need to identify TRPV6 inhibitors in conjunction with a deep mechanistic understanding of their effects on the channel activity. Here we combine cryo-electron microscopy, mutagenesis, electrophysiology and molecular dynamics modeling to decipher the molecular mechanism of TRPV6 inhibition by intracellular $Mg^{2+}$. $Mg^{2+}$ appears to bind to four, one per subunit, sites around the intracellular entrance to the TRPV6 channel pore, contributed by the negatively charged residues, D489 in the transmembrane helix S5 and D580 in S6. When bound to the D489-D580 site, $Mg^{2+}$ prevents the α-to-π transition in the middle of S6 that accompanies channel opening, thus maintaining S6 entirely α-helical, locking the channel in the closed state and inhibiting TRPV6-mediated currents. Further exploration of this inhibitory mechanism may help to develop future strategies for the treatment of TRPV6-associated diseases.

Member 6 of the vanilloid family of transient receptor potential channels, TRPV6, is a master regulator of organismal $Ca^{2+}$ homeostasis. TRPV6 channels are inherently active, continuously transitioning between conducting and non-conducting states[1,2]. The likelihood of adopting a non-conducting conformation is influenced by various regulatory factors present in the environment[1,2]. TRPV6 acts as the major dietary calcium uptake channel in the human gut[3], and as such, this channel remains constitutively active, while its activity is regulated positively by membrane lipids such as phosphatidylinositol 4,5-bisphosphate ($PIP_2$)[4–6] and negatively by $Mg^{2+}$ ions[7]. The latter regulation is speculated to contribute to the pronounced inward rectification observed for this channel[7,8]. Besides $Mg^{2+}$, calmodulin (CaM) downregulates TRPV6 by inducing $Ca^{2+}$-

dependent inactivation through highly specific protein-protein interactions[9–13].

Mutations and abnormal expression of TRPV6 have been associated with various human diseases linked to disrupted calcium balance, such as transient neonatal hyperparathyroidism, skeletal undermineralization and dysplasia, hypercalciuria, chronic pancreatitis, reproductive disorders, Pendred syndrome, and Crohn's-like disease[14–26]. Since calcium uptake plays a role in cell proliferation and cancer progression, TRPV6 is recognized as an oncochannel. Notably, TRPV6 is overexpressed in some of the most aggressive human cancers, including leukemia, as well as breast, prostate, colon, ovarian, thyroid, and endometrial cancers[27–31], and its overexpression correlates with disease severity[32,33]. Evidently, a molecular-level understanding of TRPV6 endogenous regulation by

¹Department of Biochemistry and Molecular Biophysics, Columbia University, New York, NY, USA. ²Walther-Straub Institute of Pharmacology and Toxicology, LMU Munich, Munich, Germany. ³Shemyakin-Ovchinnikov Institute of Bioorganic Chemistry, Russian Academy of Sciences, Moscow, Russia. ⁴Research Institute for Systems Biology and Medicine, Moscow, Russia. ⁵Comprehensive Pneumology Center, German Center for Lung Research, Munich, Germany. ⁶National Research University Higher School of Economics, Moscow, Russia. ⁷These authors contributed equally: Arthur Neuberger, Alexey Shalygin, Irina I. Veretenenko. ✉e-mail: vladimir.chubanov@lrz.uni-muenchen.de; efremov@nmr.ru; as4005@cumc.columbia.edu

physiological agonists and antagonists is vital to inform the much-needed drug design against the many forms of human diseases that are directly linked to channel malfunction. While multiple structural studies have successfully deciphered different types of TRPV6 inactivation or block by various exogenous antagonists[34–40], efforts to structurally characterize the endogenous regulatory factors of TRPV6 have so far been limited to inactivation by CaM[12,13].

In this study, we use a combination of single-particle cryo-electron microscopy (cryo-EM), electrophysiology and molecular modeling to decipher the mechanism of endogenous TRPV6 regulation by intracellular magnesium ions, therewith revealing how this crucial channel is regulated physiologically. We discover four $Mg^{2+}$ binding sites (one per TRPV6 subunit) surrounding the intracellular entry to the pore but not directly interfering with ion channel conductance. Binding of $Mg^{2+}$ to these sites locks TRPV6 in the closed state, thus preventing channel opening and inhibiting TRPV6-mediated currents.

## Results

### Functional characterization of human TRPV6 block by $Mg^{2+}$

To study the impact of intracellular $Mg^{2+}$ on TRPV6, we recorded whole-cell currents from HEK 293 cells expressing human TRPV6 (hTRPV6). To examine the inhibitory effect of $Mg^{2+}$, we used divalent cation-free extracellular and intracellular solutions. By applying the voltage ramp protocol (Fig. 1a), we acquired large inward and less pronounced outward hTRPV6 currents with a characteristic current-voltage (I-V)

relationship (Fig. 1b). In line with the previous work[7], the addition of 2 mM intracellular $Mg^{2+}$ caused a reduction of the inward currents recorded at negative membrane potentials and complete block of outward currents recorded at positive membrane potentials (Fig. 1b).

To get an insight into the kinetics of hTRPV6 inhibition by intracellular $Mg^{2+}$, we used a voltage step protocol (Fig. 1c). In the absence of $Mg^{2+}$, the inward and outward currents recorded in response to the voltage step from −100 to 40 mV developed immediately (Fig. 1d). However, in the presence of 2 mM intracellular $Mg^{2+}$, the increase of inward currents was significantly delayed, while the outward currents were not detectable (Fig. 1d). We concluded that the $Mg^{2+}$ block was relieved at the negative membrane potentials but remained unaffected at the positive membrane potentials.

We also examined hTRPV6 responses to voltage steps from −100 mV to the potentials in the broad range of −160 to +100 mV (Fig. 1c) and found that the resulting I-V curves obtained in the absence and presence of 2 mM intracellular $Mg^{2+}$ (Fig. 1e) were grossly similar to the ones obtained using the voltage ramp protocol (Fig. 1b). Hence, consistent with the previous reports[7,41,42], we found that intracellular $Mg^{2+}$ causes inhibition of TRPV6 currents.

### Structure of hTRPV6 in the presence of $Mg^{2+}$

To study the molecular determinants of TRPV6 inhibition by intracellular $Mg^{2+}$, we employed cryo-EM. To avoid possible interference of $Mg^{2+}$ inhibition with CaM-mediated inactivation, we utilized the

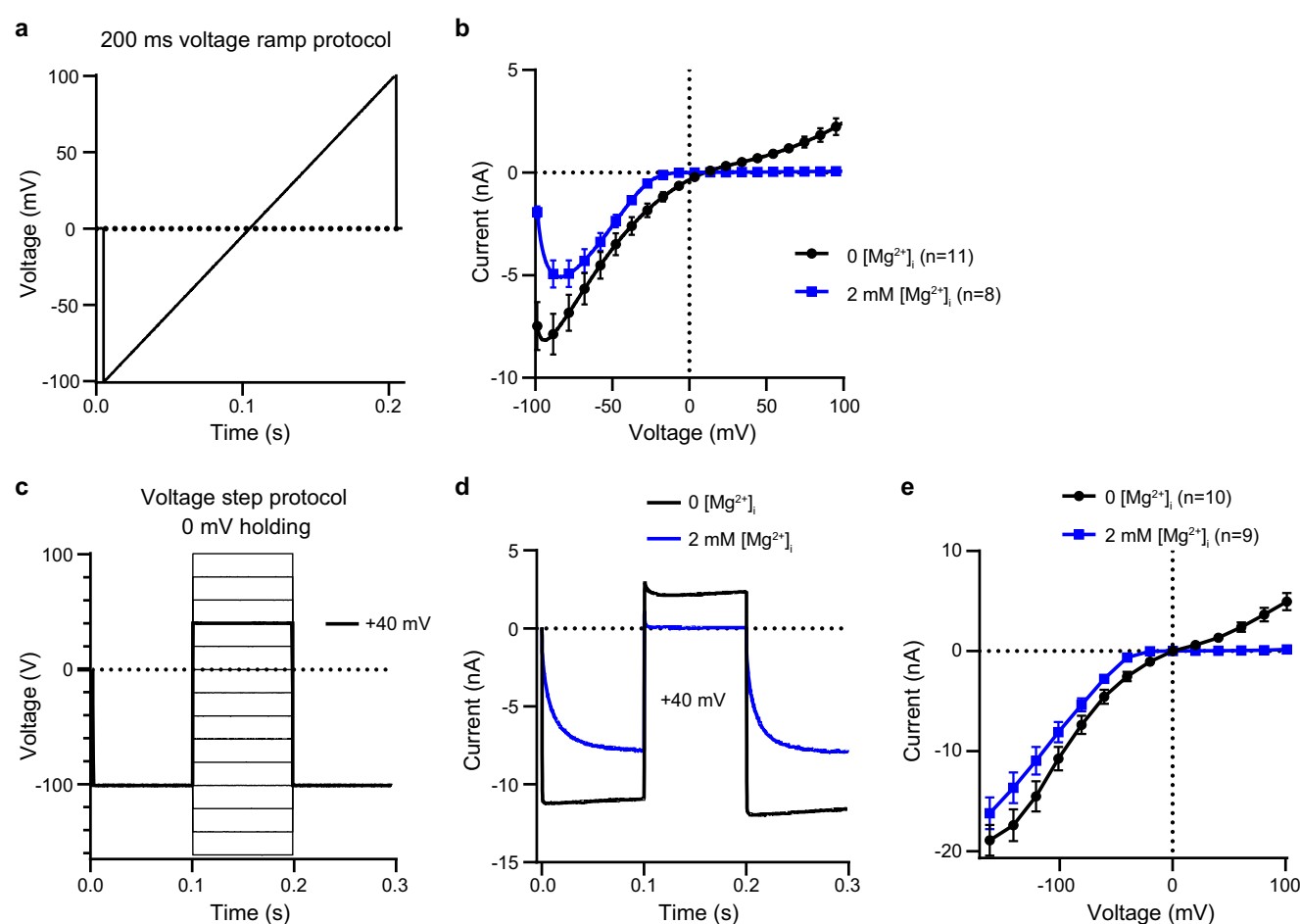

**Fig. 1 | hTRPV6 function at different concentrations of intracellular $Mg^{2+}$.**
**a** Voltage ramp protocol. **b** Average voltage dependence of whole-cell currents (mean ± SEM) recorded from hTRPV6-expressing HEK293 cells in the absence (black) or presence of 2 mM intracellular $Mg^{2+}$ (blue) using the protocol shown in (**a**). **c** Voltage step protocol. **d** Representative whole-cell currents recorded in the absence (black) or presence of 2 mM intracellular $Mg^{2+}$ (blue) using the protocol illustrated in (**c**) for the step from −100 to +40 mV. **e** Average voltage dependence of whole-cell currents (mean ± SEM) measured at the time point of 199 ms using the protocol in (**c**). $n$ is the number of cells examined.

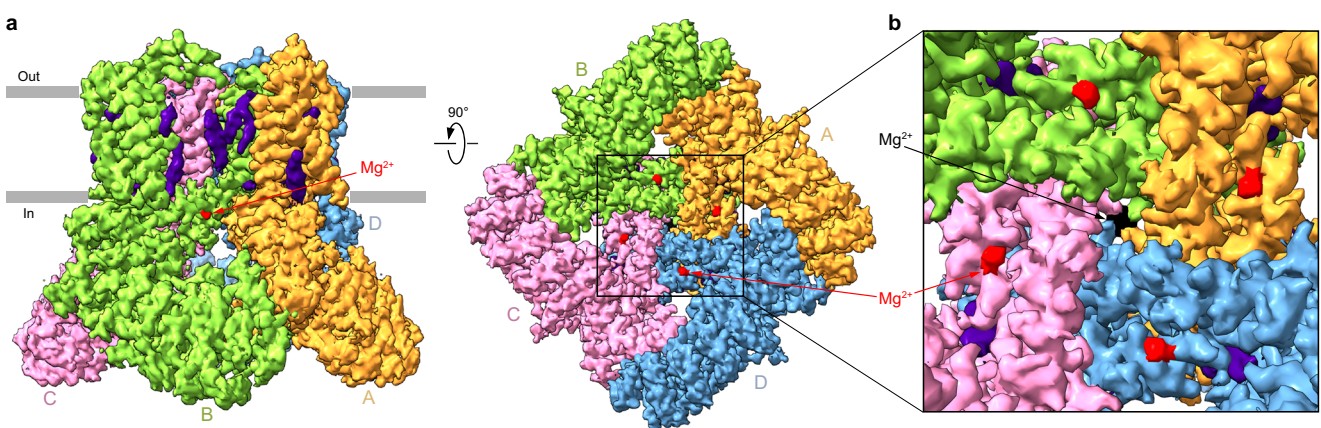

**Fig. 2 | Cryo-EM reconstruction of hTRPV6$_{Mg}$. a** 3D cryo-EM density for hTRPV6$_{Mg}$, viewed parallel to the membrane (left) and intracellularly (right), with TRPV6 subunits colored yellow, green, pink, and blue, lipids purple, and Mg$^{2+}$ ions in the selectivity filter and at the intracellular pore entrance black and red, respectively. **b** Close-up view of the region boxed in (**a**).

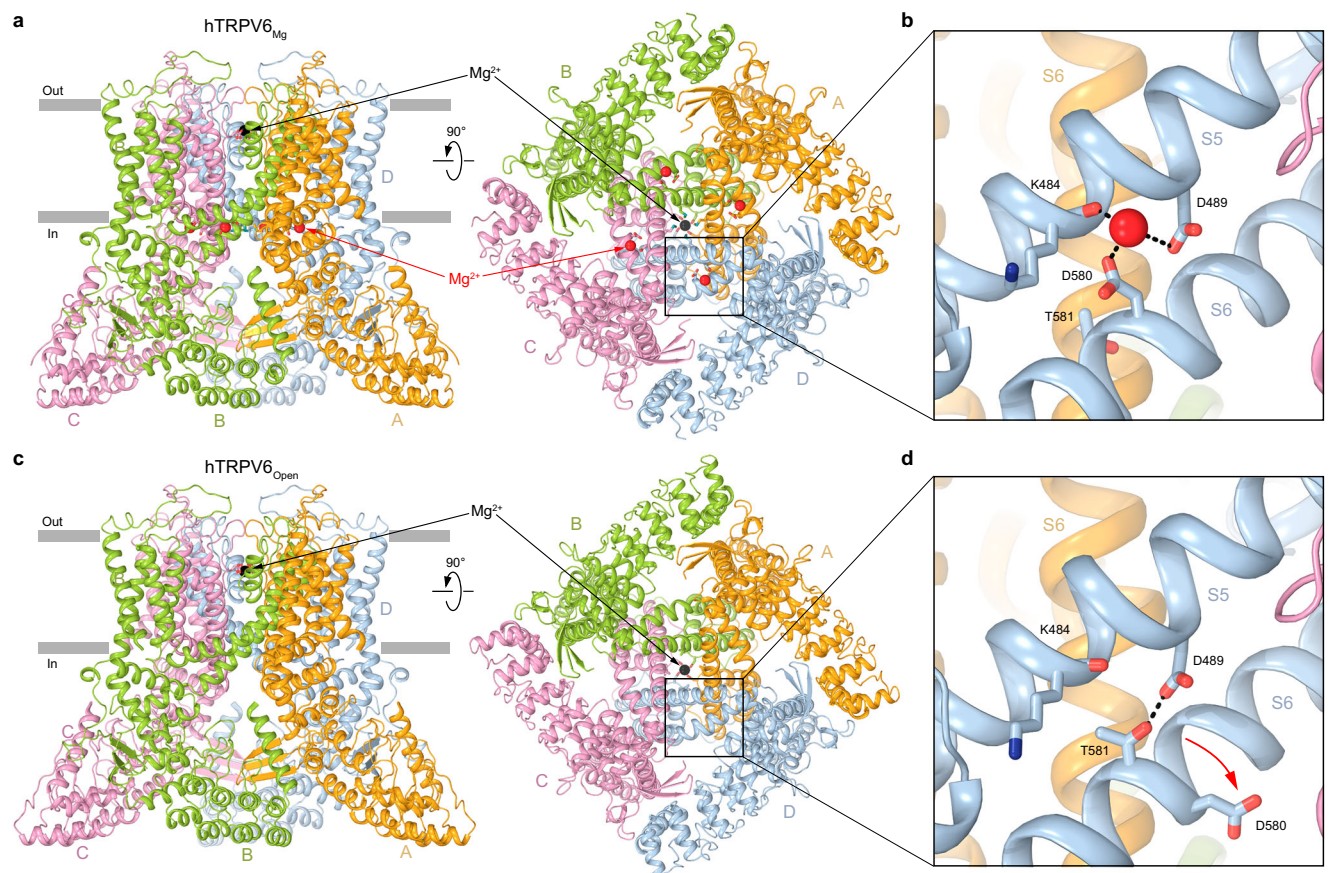

**Fig. 3 | Structure of hTRPV6$_{Mg}$ in comparison to hTRPV6$_{Open}$. a, c** Structures of hTRPV6$_{Mg}$ (**a**) and hTRPV6$_{Open}$ (7S8B) (**c**) viewed parallel to the membrane (left) and intracellularly (right), with TRPV6 subunits colored yellow, green, pink, and blue. Mg$^{2+}$ ions in the selectivity filter and at the intracellular pore entry are shown as black and red spheres, respectively. Boxed regions are expanded in (**b**) and (**d**). **b, d** Close-up views of the regions boxed in (**a**) and (**c**), with residues contributing to Mg$^{2+}$ binding shown as sticks. The red arrow in (**d**) shows rotation of S6 by ~100° that takes D580 away from the Mg$^{2+}$ binding site.

C-terminally truncated hTRPV6 construct (hTRPV6-CtD) that lacks the CaM-binding site and has been extensively used in the previous structural studies of TRPV6 as its function is identical to wild-type channels with the only exception of lacking CaM-induced inactivation[12,37–40]. Single-particle cryo-EM analysis of purified, lipid nanodisc-reconstituted hTRPV6-CtD, henceforth referred to as TRPV6, in the presence of Mg$^{2+}$ (Supplementary Figs. 1–3) yielded a 2.97-Å resolution hTRPV6$_{Mg}$ structure (Figs. 2 and 3a).

The hTRPV6$_{Mg}$ structure shares a similar overall architecture with previously determined TRPV6 structures[12,13,36–40,43–45]. It consists of four subunits and features a transmembrane domain (TMD) that houses the central ion channel pore, along with an intracellular skirt primarily composed of ankyrin repeat domains interconnected by three-stranded β-sheets, N-terminal helices, and C-terminal hooks. Amphipathic TRP helices run nearly parallel to the membrane and interact with both the TMD and the skirt. The TMD includes six transmembrane

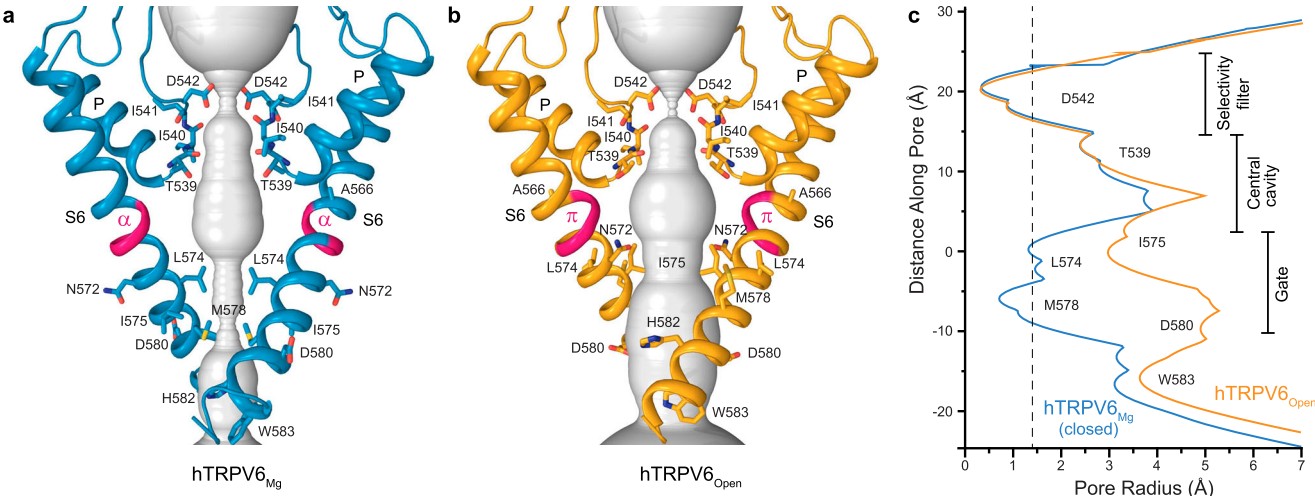

**Fig. 4 | Closed pore in hTRPV6_Mg versus open pore in hTRPV6_Open. a, b** Pore-forming domains in hTRPV6_Mg (**a**) and hTRPV6_Open (7S8B) (**b**), with residues contributing to pore lining shown as sticks. Only two of four subunits are shown, with the front and back subunits omitted for clarity. The pore profile is shown as a space-filling model (gray). The region that undergoes the α-to-π transition in S6 is highlighted in pink. **c** Pore radius for hTRPV6_Mg (blue) and hTRPV6_Open (orange) calculated using HOLE[102]. The vertical dashed line denotes the radius of a water molecule, 1.4 Å.

helices (S1–S6) and a re-entrant pore loop (P-loop) between S5 and S6. The first four transmembrane helices form the S1–S4 domain, which functions as a voltage sensor in voltage-gated ion channels[46]. In each subunit, the pore domain, assembled of S5, the P-loop, and S6, is positioned against the S1–S4 domain of the adjacent subunit in a domain-swapped configuration[44].

## Putative Mg²⁺ binding site

Close inspection of the hTRPV6_Mg cryo-EM map revealed several non-protein densities (Fig. 2a). Besides the densities for a magnesium ion tightly bound inside the channel selectivity filter and annular lipids surrounding the TMD, commonly observed in the previously published TRPV6 structures[12,13,36–40,43–45], we observed four spherical densities, one per hTRPV6_Mg subunit, around the intracellular entrance to the ion channel pore (Fig. 2b). Since these densities were not observed in the maps of hTRPV6 solved in the absence of Mg²⁺, we modeled these densities as Mg²⁺ ions (Fig. 3a). Interestingly, 3D reconstruction in C1 revealed densities for only 3 out of 4 Mg²⁺ ions (Supplementary Fig. 3b–f). While it is tempting to use these results to speculate about Mg²⁺ binding stoichiometry, one should be cautious in interpretation, given that particle classification may not be entirely accurate (small signals for Mg²⁺ ions compared to the large signal for the 4-fold symmetrical TRPV6 protein) and the resulting C1 map might represent mixed stoichiometries. From this perspective, even binding one Mg²⁺ ion per TRPV6 tetramer might be sufficient to alter the ion channel conformation.

Each Mg²⁺ ion in these putative Mg²⁺ binding sites is coordinated by the backbone carbonyl of K484 in S5 and carboxyl groups of D489 in S5 and D580 in S6 (Fig. 3b). Importantly, in the open (apo) state structure of hTRPV6 solved in the same condition but with no Mg²⁺ bound (7S8B)[40], hTRPV6_Open (Fig. 3c), the putative Mg²⁺ binding sites are absent, because the coordinating residues do not form the site anymore. Indeed, the rotation of S6 takes the side chain of D580 away from K484 and D489 (Fig. 3d), explaining the lack of the corresponding density in the cryo-EM map of hTRPV6_Open[40].

We further inspected whether hTRPV6_Open and hTRPV6_Mg can form other potential Mg²⁺ binding sites using molecular modeling. The search criteria were based on the spatial proximity of negatively charged protein groups capable of coordinating Mg²⁺ on the cytoplasmic side of the TMD and the presence in these locations of well-defined cryo-EM density. Three potential Mg²⁺ binding sites on the

intracellular side of the TMD were identified in hTRPV6_Mg (Supplementary Fig. 4): (1) the D489-D580 site between S5 and S6, (2) the D309-E588-D590-E591 site between the S6-TRP helix kink and the LH2 helix, and (3) the E282-D284-D288-E289-E294-E605 site between the LH1 helix and the C-terminus of the TRP helix. The D489-D580 site is only present in hTRPV6_Mg because of the S6 helix rotation, whereas the other two sites are present in both hTRPV6_Mg and hTRPV6_Open, and Mg²⁺ binding to these sites does not appear to alter the equilibrium between two states. Combined with the presence of pronounced cryo-EM density in hTRPV6_Mg (Fig. 2), this suggests the potential involvement of the D489-D580 site in TRPV6 inhibition by intracellular Mg²⁺.

## hTRPV6_Mg structure represents the closed state

Given the rotation of S6 that correlates with the loss of the putative Mg²⁺ binding site in hTRPV6_Open versus hTRPV6_Mg (Fig. 3), we compared the channel gating conformations in these two states (Fig. 4). The pore in the hTRPV6_Mg structure has two narrow constrictions, one at the selectivity filter, formed by the calcium-coordinating side chains of D542, and the second at the gate region, formed by the hydrophobic side chains of L574 and M578 (Fig. 4a). This pore conformation is typical for the closed-state TRPV6 structures and different from the open, apo-state structures. Indeed, while the selectivity filter in hTRPV6_Open is nearly identical to the one in hTRPV6_Mg, the gate region is much wider, with the narrow constriction formed by the side chains of isoleucines I575 (Fig. 4b).

Typical for the open-to-closed transition during TRPV6 gating[45], the narrowing of the hTRPV6_Mg pore is accompanied by a ~100-degree rotation of the S6 intracellular portion (Fig. 4c). During this rotation, S6, which has a π-bulge following the gating hinge alanine A566 in hTRPV6_Open, becomes entirely α-helical in hTRPV6_Mg. As a result, a completely different set of residues faces the pore in the closed and open states, with the narrow constriction contributed by I575 in hTRPV6_Open and L574 and M578 in hTRPV6_Mg (Fig. 4a, b). Importantly, the open-to-closed state transition is accompanied by the loss of electrostatic interactions between Q473 in the S4-S5 linker and R589 in the TRP helix as well as D489 in S5 and T581 in S6, which support the energetically unfavorable α-to-π transition in S6 and the open state, accordingly[45]. Interestingly, the second type of open state-stabilizing interactions involve D489, which contributes to the coordination of the putative Mg²⁺ ions in hTRPV6_Mg (Fig. 3a, b). By recruiting D489 and D580 into coordination, Mg²⁺ ions, therefore, disfavor the interaction

between D489 and T581 (Fig. 3b, d) and stabilize the closed state, accordingly.

## Functional effects of mutations at the putative Mg$^{2+}$ binding site

To test the contribution of the putative Mg$^{2+}$ binding site discovered in hTRPV6$_{Mg}$ to the channel function and inhibition by Mg$^{2+}$, we made mutations of the residues D489 and D580, which contribute their side chains to the predicted coordination of Mg$^{2+}$ ions (Fig. 3b). By applying voltage ramp protocol, we examined how these mutations affect TRPV6 currents in the presence of 2 mM intracellular Mg$^{2+}$ (Fig. 5a). We found that the amplitudes of currents recorded from HEK 293 cells expressing D489A, D489N and D489E hTRPV6 channel mutants were indistinguishable from the amplitudes of background currents recorded from untransfected (control) HEK 293 cells, suggesting that the D489A, D489N and D489E mutations resulted in the loss of the TRPV6 channel function (Fig. 5b). However, the D580E, D580R, D580N and D580A TRPV6 mutants retained channel activity (Fig. 5a) and, therefore, were selected for further functional assessment using the voltage step protocol (Fig. 5c–g). Specifically, we investigated whether the D580E, D580R, D580N, and D580A substitutions could affect the ability of 2 mM intracellular Mg$^{2+}$ to delay the development of inward currents at the step from +60 mV to −100 mV and found that the D580R mutation caused the most remarkable changes in current relaxation (Fig. 5c–g). Further analysis of outward currents at +60 mV revealed that the D580R, D580N, and D580A mutations mitigated the Mg$^{2+}$ block of outward currents, whereas the D580E substitution did not impact this channel characteristic (Fig. 5h, i), suggesting that the negatively charged side chain D580 is crucial for Mg$^{2+}$ effects on hTRPV6.

Because the D580R mutation caused the most remarkable changes in TRPV6-mediated currents, this mutant was selected for further functional analysis. We used three approaches to compare the responses of WT and D580R TRPV6 channels to 0.5, 2 and 10 mM intracellular Mg$^{2+}$ (Fig. 6). First, using the voltage ramp protocol, we found that the D580R mutation significantly reduced the influence of 0.5 and 2 mM Mg$^{2+}$ on the current amplitude, especially at positive membrane potentials (Fig. 6a, b). In contrast, the inhibition of TRPV6-D580R currents by 10 mM intracellular Mg$^{2+}$ remained strong, similar to wild-type channels (Fig. 6a, b). Second, whole-cell recordings made using the voltage steps protocol, in which the −100 mV membrane potential pre-pulse was followed by the membrane potentials steps ranging from −160 to +100 mV, reinforced the idea that D580R strongly affected the sensitivity of the channel to 0.5 and 2 mM Mg$^{2+}$, with the most substantial impact on currents in the range of −50 to +60 mV (Fig. 6c, d). These results suggest that the D580R mutation impaired the Mg$^{2+}$ block of TRPV6 currents by 0.5 and 2 mM Mg$^{2+}$, matching the range of physiological cytosolic concentrations of free Mg$^{2+}$ in mammalian cells[47]. However, D580R did not alter the channel responses to 10 mM Mg$^{2+}$, suggesting that high, non-physiological concentrations of Mg$^{2+}$ may act on TRPV6 through an alternative mechanism. Third, we examined the WT and D580R hTRPV6 channels using the inside-out patch-clamp recordings (Supplementary Fig. 5). Using the ramp protocol, we found that the WT channel exhibited the characteristic inward rectification, analogous to the one observed previously[7] (Supplementary Fig. 5a). In response to acute application of 2 mM Mg$^{2+}$ from the cytosolic side of the patch, the amplitude of this current was substantially reduced (Supplementary Fig. 5a). In the step protocol, WT hTRPV6-mediated inward and outward currents recorded in the absence of Mg$^{2+}$ (Supplementary Fig. 5a) had fast kinetics, similar to whole-cell currents (Fig. 1d). In the presence of 2 mM Mg$^{2+}$, the development of the WT hTRPV6-mediated inward currents was significantly delayed, while the outward currents remained blocked (Supplementary Fig. 5a). The D580R mutation altered the channel's response to Mg$^{2+}$ application (Supplementary Fig. 5b), resembling our findings for

whole-cell currents (Figs. 5 and 6). Altogether, these results support the notion that D580 contributes to the regulation of the hTRPV6 channel by intracellular Mg$^{2+}$.

## Free energy calculations for Mg$^{2+}$ binding to the D489-D580 site

To establish the characteristics of Mg$^{2+}$ interactions with the D489-D580 site, we calculated the standard binding free energy at zero ionic strength ($\Delta G^0$) using molecular dynamics simulations. To evaluate the specificity of the site, the same calculations were performed for another common divalent cation, Ca$^{2+}$. We employed the electronic continuum correction (ECC) approach, which ensures better agreement with experimentally measured binding energy compared to the "standard" force field parameterization[48]. A free energy surface (FES) was obtained by well-tempered metadynamics calculations (WTMetaD)[49] in "multiple walkers" configuration[50]. The model systems were composed of hTRPV6$_{Mg}$ fragments of S5 and S6 helices containing the D489-D580 site and Mg$^{2+}$ or Ca$^{2+}$ cations (Supplementary Fig. 6a). The WTMetaD simulations were performed with three collective variables (CVs): two distances between the carbon atoms of the D489/D580 carboxyl groups and the cation ($L_1/L_2$), and the coordination number of water molecules in the first cation hydration sphere (CN$_W$).

Mg$^{2+}$ and Ca$^{2+}$ exhibited similar binding pathways through several free energy local minima, corresponding to cation approaching from bulk water (State 1), water-mediated binding to two carboxyl groups (State 2), direct binding to one carboxyl group and water-mediated binding to the other (State 3), and finally the most energetically preferable state with direct binding to both residues (State 4) (Fig. 7a, b). In State 4, each carboxyl group can bind a cation in one of two distinct modes: monodentate, when one carboxyl oxygen replaces one water molecule in the cation hydration shell, or bidentate, when two carboxyl oxygens replace two water molecules. For Mg$^{2+}$, monodentate-monodentate mode is preferable, yielding the global energy minima at two water molecules replaced in total (corresponds to CN$_W$ = 4 in Supplementary Fig. 6d, f). In contrast, Ca$^{2+}$ predominantly adopted a monodentate-bidentate binding mode, replacing three water molecules (CN$_W$ = 4 in Supplementary Fig. 6e, g). The distances between Mg$^{2+}$ or Ca$^{2+}$ and carboxyl oxygens (0.21 or 0.23 nm, correspondingly) are in good agreement with the quantum chemical calculations of the complexes of two acetate ions with hydrated cations. Notably, the minimum free energy paths reveal a ~10 kJ/mol higher energy barrier for Mg$^{2+}$ binding compared to Ca$^{2+}$ (Fig. 7c, d), reflecting the greater stability of the Mg$^{2+}$ hydration shell.

The standard binding free energies revealed a striking difference between Mg$^{2+}$ ($\Delta G^0 = -12.9 \pm 0.4$ kJ/mol) and Ca$^{2+}$ ($\Delta G^0 = -3.7 \pm 0.6$ kJ/mol) at the D489-D580 site. This corresponds to a ~35-fold stronger binding preference for Mg$^{2+}$ over Ca$^{2+}$, demonstrating high selectivity of the D489-D580 site for magnesium. To evaluate the structural basis of the resulting selectivity, we performed the same WTMetaD simulations for the econazole-inhibited hTRPV6 structure (TRPV6$_{Eco}$, 7S8C)[40], where the D489-D580 site is presented in magnesium-free conditions and where the distance between D489 and D580 C$\alpha$ atoms is 0.1 nm larger than in TRPV6$_{Mg}$ (Supplementary Fig. 7a). The value of $\Delta G^0$ was found to be $-9.4 \pm 0.5$ kJ/mol for Mg$^{2+}$ and $-4.8 \pm 0.4$ kJ/mol for Ca$^{2+}$ (Supplementary Fig. 7 and Supplementary Table 1). Strikingly, this tiny expansion of the site in TRPV6$_{Eco}$ significantly weakened the Mg$^{2+}$ binding affinity ($\Delta\Delta G^0 = +3.5 \pm 0.6$ kJ/mol in TRPV6$_{Eco}$ versus TRPV6$_{Mg}$), while Ca$^{2+}$ binding remained essentially unaffected ($\Delta\Delta G^0 = -1.1 \pm 0.7$ kJ/mol). This differential effect arises from distinct ion coordination chemistry: in Mg$^{2+}$-optimized geometry of TRPV6$_{Mg}$, the arrangement of D489 and D580 oxygen atoms stabilizes a near-perfect octahedral first hydration shell around Mg$^{2+}$, whereas site expansion disrupts this configuration, reducing the Mg$^{2+}$ affinity. Conversely, the larger and more flexible coordination sphere of Ca$^{2+}$

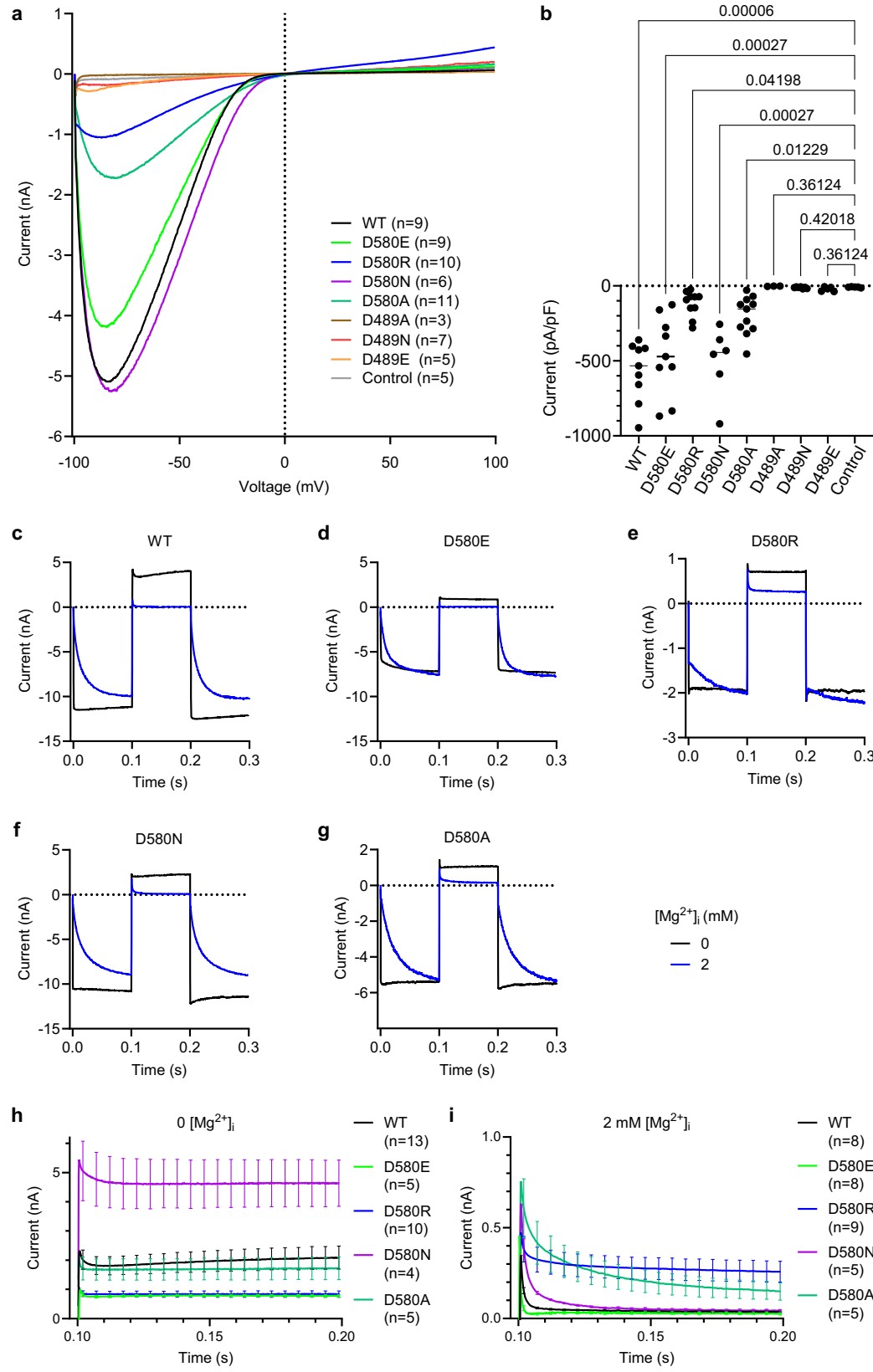

**Fig. 5 | Functional characterization of hTRPV6 mutants. a** Representative whole-cell currents recorded from HEK 293 cells expressing wild-type (WT) or mutant hTRPV6 at 2 mM intracellular Mg²⁺ using the voltage ramp protocol illustrated in Fig. 1a. The data for WT currents are from Fig. 1b. Control represents non-transfected HEK 293 cells; n is the number of cells examined. **b** Current amplitudes measured at −80 mV using currents illustrated in (**a**) and normalized by cell size (mean ± SEM). *P*-values were determined using the Kruskal–Wallis test with multiple comparison corrections by Benjamini, Krieger, and Yekutieli test. Source data

are provided. **c**–**g** Representative currents recorded from HEK 293 cells expressing wild-type TRPV6 (**c**) or mutants D580E (**d**), D580R (**e**), D580N (**f**), and D580A (**g**) in the absence (black) or presence of 2 mM intracellular Mg²⁺ (blue) using the step protocol similar to illustrated in Fig. 1c, with 0 mV holding membrane potential and the voltage step from −100 to +60 mV. **h, i** Average amplitude of currents (mean ± SEM) recorded at +60 mV in the absence (**h**) or presence of 2 mM Mg²⁺ (**i**) using the protocols illustrated in (**c**–**g**). *n* is the number of cells examined.

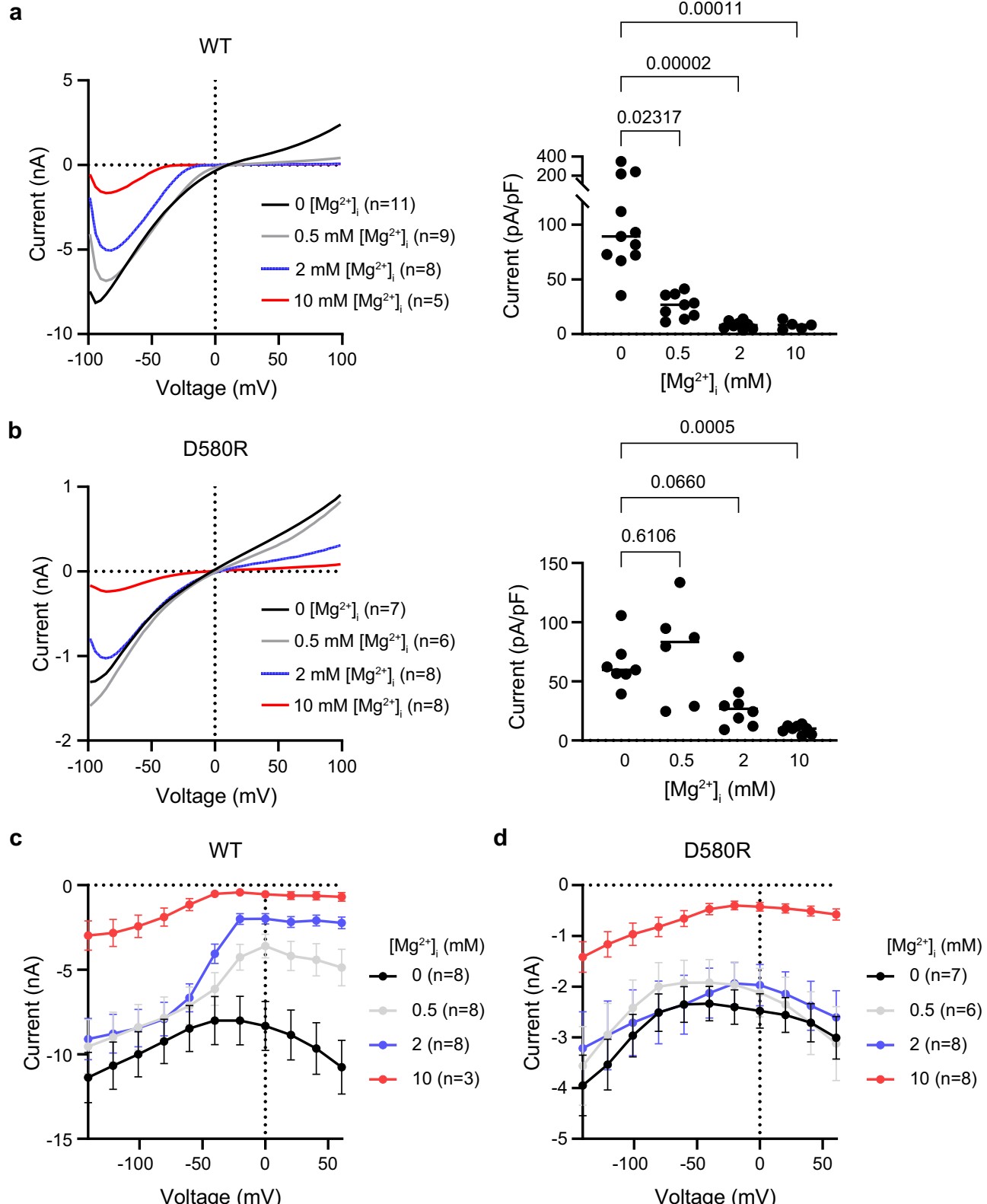

**Fig. 6 | Effect of D580R mutation on current inhibition by intracellular Mg²⁺.**
**a**, **b** On the left, the average voltage dependence of whole-cell currents recorded from HEK 293 cells expressing wild-type hTRPV6 (**a**) and D580R (**b**) in the absence or presence of different concentrations of intracellular Mg²⁺ using the voltage ramp protocol illustrated in Fig. 1a. The data for WT currents at 0 and 2 mM Mg²⁺ were taken from Fig. 1b, the data for D580R at 2 mM Mg²⁺ were taken from Fig. 5a. On the right, the current amplitudes normalized by cell size (mean ± SEM) measured at +80 mV using the currents on the left. *P*-values were determined using the Kruskal–Wallis test with multiple comparison corrections by Benjamini, Krieger, and Yekutieli test. Source data are provided. **c**, **d** Initial tail current amplitudes (mean ± SEM) measured for wild-type TRPV6 (**c**) and D580R (**d**) after returning to −100 mV during the voltage step protocol illustrated in Fig. 1c at different concentrations of intracellular Mg²⁺. *n* is the number of cells examined.

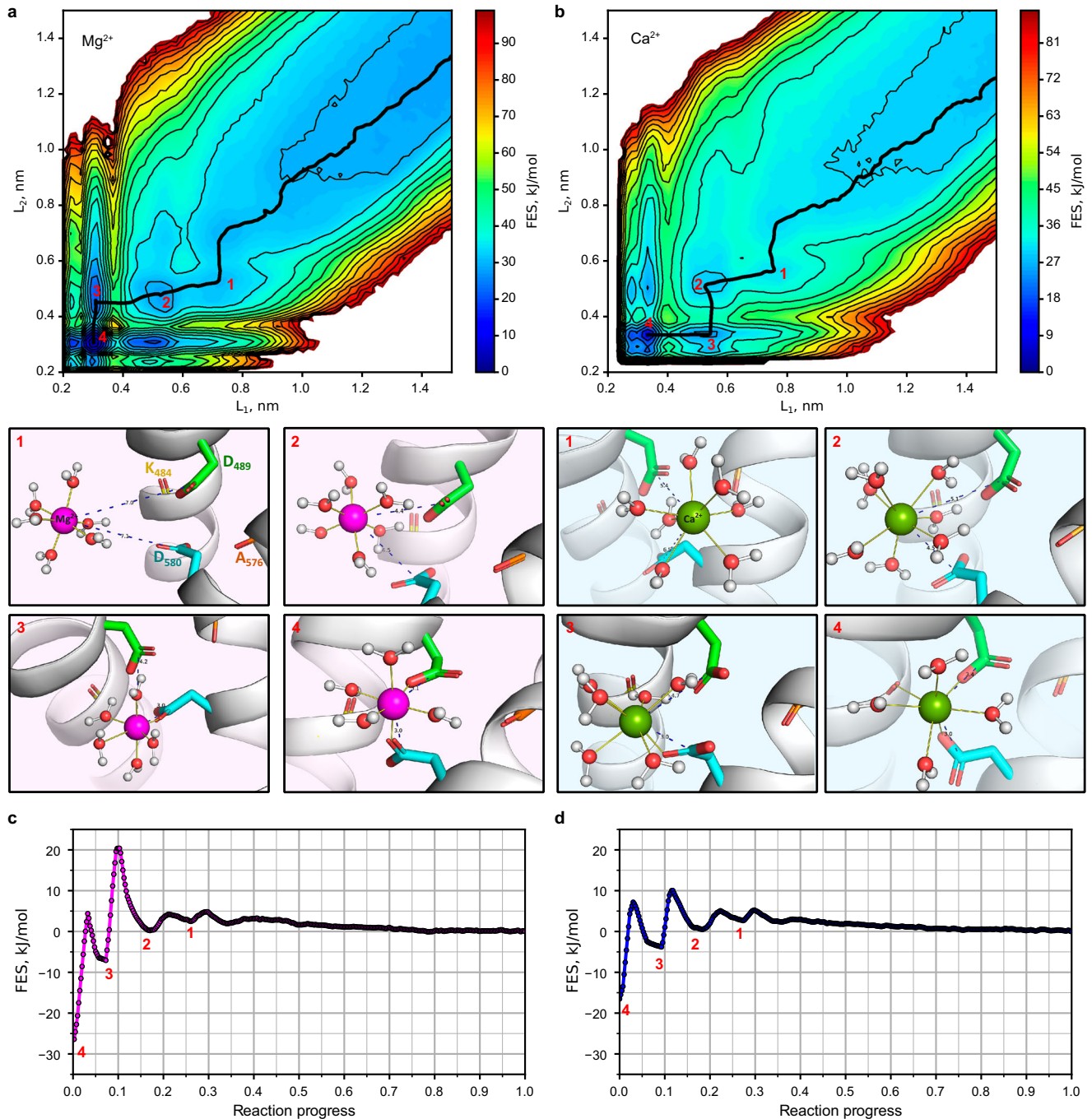

**Fig. 7 | Modeling of Mg²⁺ and Ca²⁺ binding to the D489-D580 site in TRPV6_Mg.**
**a**, **b** 2D projection of the free energy surface (FES) to L1 (distance between the cation and the Cγ atom of D489) and L2 (distance between the cation and the Cγ atom of D580) collective variables. Black line shows the minimum free energy path (MFEP). Lower inserts represent the states corresponding to the crucial local

minima of MFEP: (1) the cation approaching from bulk, (2) water-mediated binding of the cation to both D489 and D580 carboxyl groups, (3) direct cation binding to one carboxyl group and water-mediated binding to another, and (4) direct cation binding to both D489 and D580. **c**, **d** Free energy profile along the MFEP. Red numbers indicate the states illustrated in (**a** and **b**).

exhibits lower sensitivity to such geometric perturbation. These results demonstrate a crucial role of precise site geometry in Mg²⁺ selectivity and support the idea that the conformation of TRPV6_Mg was evolutionary optimized to bind Mg²⁺ ions.

## Discussion

Multiple endogenous factors, such as protons, lipids, nucleotides, Ca²⁺ and Mg²⁺, tightly regulate the function of TRP channels. While the structural mechanisms underlying the effects of lipids, Ca²⁺ and some other ligands are relatively well-investigated, regulation by

intracellular Mg²⁺ has remained enigmatic. TRPV6 is a highly selective Ca²⁺ channel that controls calcium transport in epithelial cells of the placenta, intestine, and other organs[28,51]. Upregulation of TRPV6 activity was also proposed to play a crucial role in tumor progression[30]. Early electrophysiological experiments demonstrated that the channel activity of TRPV6 is inhibited by intracellular Ca²⁺ acting on the channel through CaM[11,52]. In addition, inhibition of TRPV6 by intracellular Mg²⁺ contributes to the characteristic inward rectification of TRPV6-mediated currents[7,8,42]. However, given the high Ca²⁺ permeability of TRPV6 and the elaborate downregulation of TRPV6 through Ca²⁺-

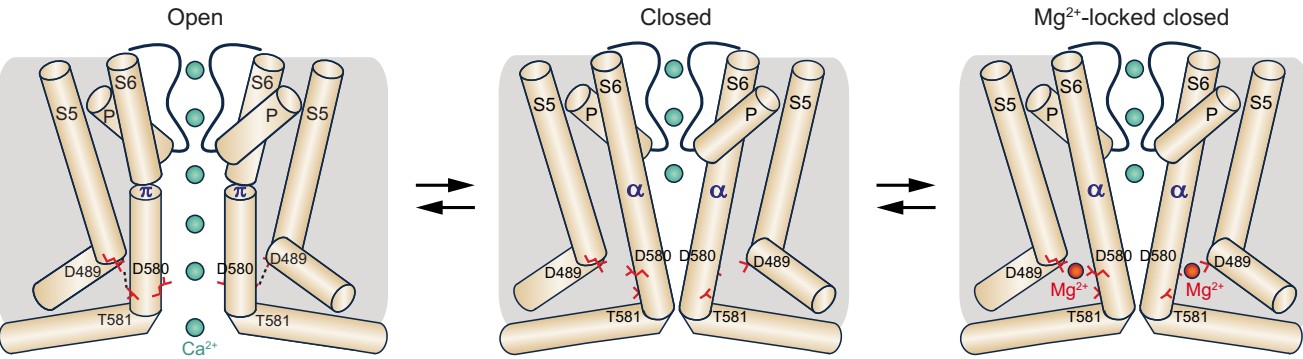

**Fig. 8 | Mechanism of TRPV6 inhibition by Mg²⁺.** Cartoons represent TRPV6 in the open, closed, and Mg²⁺-inhibited closed states. Transition from the closed to open state, stabilized by the formation of D489-T581 hydrogen bonds (dashed lines), leads to permeation of ions (green spheres) and is accompanied by a local α-to-π helical transition in S6 that maintains the selectivity filter conformation, while the lower part of S6 bends by ~11° and rotates by ~100°. Residues D489 and D580, which become distal in the open state, reside near each other in the closed state and can coordinate a Mg²⁺ ion. Mg²⁺ binding to the D489-D580 site prevents separation of these residues and rotation of the lower part of S6 that accompanies channel opening, thus locking the channel in the closed state.

bound CaM[12,13], it has remained unclear which mechanism Mg²⁺ can possibly utilize to contribute to this already complex regulation of TRPV6. Earlier electrophysiological studies proposed that intracellular Mg²⁺ inhibits TRPV6 by blocking its pore at the extracellular entry to the selectivity filter[7]. However, this model still awaits structural verification. Here, by combining single-particle cryo-EM, mutagenesis, electrophysiology, and molecular dynamics modeling, we unravel an unexpected mechanism of TRPV6 regulation by intracellular Mg²⁺ through binding of these ions to a distinct regulatory site on the intracellular side of the ion channel.

In line with the previous reports[7], our electrophysiological analysis showed that intracellular Mg²⁺ significantly reduced TRPV6 currents at negative membrane potentials and completely blocked outward currents at positive potentials, reinforcing the idea that intracellular Mg²⁺ serves as a negative regulator of the TRPV6 channel. Our structural experiments in conjunction with MD simulations demonstrated that the most probable binding site of intracellular Mg²⁺ is contributed by two acidic residues, D489 in S5 and D580 in S6. In good agreement with the structural results, mutagenesis of D580 in TRPV6, especially D580R but not D580E, mitigated the inhibitory effects of Mg²⁺ on TRPV6 currents. These findings support our mechanistic models, implying that Mg²⁺ directly interacts with and locks the TRPV6 channel in the closed nonconducting state.

It is important to note that the D489-D580 site is present only when the channel is in the closed state, which is characterized as an obligatory non-conducting state with the entirely α-helical S6[53]. Other potential Mg²⁺ binding sites at the intracellular side of the TMD are present in both open and closed states of TRPV6, and ion binding to these sites does not appear to alter the protein conformation. In the open state, which is stabilized by the hydrogen bond between D489 and T581 and has a π-bulge in the middle of S6, the carboxyl groups of D489 and D580 are too far apart and can be brought close to each other due to the open-to-closed state transition that is accompanied by ~100° rotation of the C-terminal part of S6 and a π-to-α transition in the middle of this helix (Fig. 8). When D489 and D580 are nearby, they can coordinate a Mg²⁺ ion, which prevents the reverse transition from the closed to open state. Thus, Mg²⁺ hijacks D489 from its open-state stabilizing interaction with T581 and locks the channel in the closed state, leading to inhibition of TRPV6-mediated currents.

Our results, therefore, shed light on the physiological modulation of TRPV6 currents by intracellular Mg²⁺ via binding of this cation to a distinct site. The physiological role of this regulatory mechanism is a subject of future studies. However, in analogy to the situation with other TRP channels, TRPM6 and TRPM7[54-56], it is possible that, apart from calmodulin, Mg²⁺ mediates an alternative feedback mechanism,

which is necessary to prevent excessive activity of TRPV6 leading to Ca²⁺ overload. Hence, the identified Mg²⁺ binding site can potentially be explored for the design of drugs to downregulate TRPV6 in disease conditions.

To our knowledge, the identified regulatory site of TRPV6 has not been discovered in TRP channels before[34,35]. It is worth noting that several other TRPV channels, including TRPV1, TRPV2, TRPV3, and TRPV5, are known to be modulated by intracellular Mg²⁺ ions[57-60]. Mechanistically, however, such regulatory effects of Mg²⁺ on TRPV channels remain poorly understood. To this end, the present study offers a structural insight into the role of Mg²⁺ in the inhibition of TRPV6 currents, and future studies are necessary to define whether a similar mechanism is conserved among TRPV channels.

## Methods
### Constructs and cell lines
C-terminally truncated human TRPV6 (hTRPV6-CtD, residues 1–666 of wild-type channel) used in the previous cryo-EM studies of the hTRPV6 channel[37] was cloned into a pEG BacMam vector[61] with a C-terminal thrombin cleavage site followed by a streptavidin affinity tag (WSHPQFEK). Point mutations in wild-type human TRPV6 were introduced using the standard molecular biology techniques as described before[38-40,62-66].

For structural experiments, expression of human TRPV6 was performed in HEK 293S cells lacking N-acetyl-glucosaminyltransferase I (GnTI⁻, mycoplasma test negative, ATCC #CRL-3022) that were maintained at 37 °C and 6% CO₂ in Freestyle 293 medium (Thermo Fisher Scientific #12338-018) supplemented with 2% FBS (Thermo Fisher Scientific, #10270106). Baculovirus for infecting HEK 293S GnTI⁻ cells was produced in Sf9 cells (GIBCO) cultured in the Sf-900 III SFM media (GIBCO) at 27 °C. For patch-clamp experiments, TRPV6 channels were expressed in HEK 293T cells (mycoplasma test negative, ATCC #CRL3216) that were maintained at 37 °C and 5% CO₂ in DMEM (Merck, #D6429) supplemented with 10% FBS, 100 μg/ml streptomycin and 100 U/ml penicillin (Merck, #P4333).

### Expression and purification
hTRPV6 was expressed and purified using our previously established protocols[38-40,63-66]. Bacmids and baculoviruses were produced using the standard procedures[61,63]. Baculovirus was made in Sf9 cells for ~72 h (Thermo Fisher Scientific, mycoplasma test negative, GIBCO #12659017) and was added to suspension-adapted HEK 293S cells lacking N-acetyl-glucosaminyltransferase I (GnTI⁻, mycoplasma test negative, ATCC #CRL-3022) that were maintained in Freestyle 293 media (Gibco-Life Technologies #12338-018) supplemented with 2%

FBS at 37 °C and 5% $CO_2$. Twenty-four hours after transduction, 10 mM sodium butyrate was added to the cells to enhance protein expression, and the temperature was reduced to 30 °C. Seventy-two hours after transduction, the cells were harvested by centrifugation at 5471 × $g$ for 15 min using a Sorvall Evolution RC centrifuge (Thermo Fisher Scientific), washed in phosphate-buffered saline pH 8.0, and pelleted by centrifugation at 3202 × $g$ for 10 min using an Eppendorf 5810 centrifuge. The cell pellet was solubilized under constant stirring for 2 h at 4 °C in ice-cold lysis buffer containing 1% (w/v) n-dodecyl β-D-maltoside, 0.1% (w/v) CHS, 20 mM Tris-Cl pH 8.0, 150 mM NaCl, 0.8 µM aprotinin, 4.3 µM leupeptin, 2 µM pepstatin A, 1 mM phenylmethylsulfonyl fluoride, and 1 mM β-mercaptoethanol (βME). The non-solubilized material was pelleted in the Eppendorf 5810 centrifuge at 3202 × $g$ and 4 °C for 10 min. The supernatant was subjected to ultracentrifugation in a Beckman Coulter ultracentrifuge using a Beckman Coulter Type 45Ti rotor at 186,000 × $g$ and 4 °C for 1 h to further clean up the solubilized protein. The supernatant was added to a strep resin and rotated for 14–16 h at 4 °C. The resin was washed with 10 column volumes of the wash buffer containing 20 mM Tris-HCl pH 8.0, 150 mM NaCl, 1 mM βME, 0.01% (w/v) GDN, and 0.001% (w/v) CHS, and the protein was eluted with the same buffer supplemented with 2.5 mM D-desthiobiotin. The eluted protein was concentrated using a 100 kDa NMWL centrifugal filter (MilliporeSigma Amicon) to 0.5 ml and then centrifuged in a Sorvall MTX 150 Micro-Ultracentrifuge (Thermo Fisher Scientific) using an S100AT4 rotor for 30 min at 66,000 × $g$ and 4 °C before injection into a size-exclusion chromatography (SEC) column. hTRPV6 was further purified using a Superose™ 6 10/300 GL SEC column attached to an AKTA FPLC (GE Healthcare) and equilibrated in 150 mM NaCl, 20 mM Tris-HCl pH 8.0, 1 mM βME, 0.01% GDN, and 0.001% CHS. The tetrameric peak fractions were pooled and concentrated using 100-kDa NMWL centrifugal filter to ~3 mg/ml.

hTRPV6 was reconstituted into circularized NW11 nanodiscs (cNW11). cNW11 nanodiscs were prepared according to the standard protocol[66–68] and stored before usage at −80 °C as ~2–3-mg/ml aliquots in the buffer containing 20 mM Tris pH 8.0 and 150 mM NaCl. For nanodisc reconstitution, the purified protein was mixed with cNW11 nanodiscs and soybean lipids (Soy polar extract, Avanti Polar Lipids) at the molar ratio of 1:3:166 (hTRPV6:cNW11:lipids). The lipids were dissolved in the buffer containing 150 mM NaCl and 20 mM Tris pH 8.0 to reach the concentration of 100 mg/ml and subjected to 5–10 cycles of freezing in liquid nitrogen and thawing in a water bath sonicator. The nanodisc mixture (500 µl) was rocked at room temperature for 1 h. Subsequently, 40 mg of Bio-beads SM2 (Bio-Rad), pre-wet in the buffer containing 20 mM Tris pH 8.0 and 150 mM NaCl, were added to the nanodisc mixture, which was then rotated for one hour at 4 °C. After adding 40 mg more of Bio-beads SM2, the resulting mixture was rotated at 4 °C for another ~14 h. The Bio-beads SM2 were then removed by pipetting. The sample was then centrifuged in a Sorvall MTX 150 Micro-Ultracentrifuge (Thermo Fisher Scientific) using a S100AT4 rotor for 30 min at 66,000 × $g$ and 4 °C before injecting into the SEC column. Nanodisc-reconstituted hTRPV6 was then purified from empty nanodiscs using the Superose™ 6 10/300 GL SEC column equilibrated with the buffer containing 150 mM NaCl, 20 mM Tris pH 8.0, and 1 mM βME. Fractions of nanodisc-reconstituted hTRPV6 were pooled and concentrated to 1.9 mg/ml using a 100-kDa NMWL centrifugal filter. $Mg^{2+}$ was added to TRPV6 at 2 mM final concentration and incubated for 1 h on ice before freezing the grids.

### Cryo-EM sample preparation and data collection
UltrAuFoil R 1.2/1.3, Au 300 grids were used for plunge-freezing. Prior to sample application, grids were plasma treated in a PELCO easiGlow glow discharge cleaning system (0.39 mBar, 15 mA, "glow" 25 s, "hold" 10 s). A Mark IV Vitrobot (Thermo Fisher Scientific) set to 100% humidity at 4 °C was used to plunge-freeze the grids into liquid ethane

after applying 3 µl of protein sample to their gold-coated side using the blot time of 5 s, the blot force of 5, and the wait time of 15 s. The grids were stored in liquid nitrogen before imaging. Images of frozen-hydrated particles of cNW11-reconstituted TRPV6Mg were collected on a Titan Krios TEM operating at 300 kV with a post-column GIF Quantum energy filter of 20 eV and a Gatan K3 Summit DED camera using SerialEM. 5706 micrographs were collected in super-resolution mode with an image pixel size of 0.413 Å across a defocus range of −0.8 to −2.0 µm. The total dose of ~60 e⁻ Å⁻² was attained by using a dose rate of ~16 e⁻ pixel⁻¹ s⁻¹ across 50 frames for a 2.0-s total exposure time.

### Image processing and 3D reconstruction
Data were processed in RELION[69] (Supplementary Figs. 1, 2). 5706 movie frames were collected and subsequently aligned using RELION-embedded MotionCor2. Contrast transfer function (CTF) estimation was performed on non-dose-weighted micrographs using the patch CTF estimation. Subsequent data processing was done on dose-weighted micrographs. Following CTF estimation, micrographs were manually inspected and those with outliers in defocus values, ice thickness, and astigmatism as well as micrographs with lower predicted CTF-correlated resolution (lower than 5 Å) were excluded from further processing (individually assessed for each parameter relative to the overall distribution). Autopicked particles were subjected to several rounds of 2D classification and 2D-class selection to generate 2D templates for template-based autopicking in RELION. After template-based picking, 5,057,951 particle images (4x-binned) were used for 3D autorefine in C1 and iterative rounds of 3D classification followed by CTF refinement, 3D auto-refinement in C1 and Bayesian Polishing, before the best class of 31,296 particles (72% of particles in final 3D classification) was 3D auto-refined in C4 symmetry and with a soft mask yielding a 3.46 Å map that was postprocessed to a final map at 2.97 Å. This map was used for model building.

The reported resolution of 2.97 Å (Table 1) was estimated using the gold standard Fourier shell correlation (GSFSC). The local resolution was calculated with the resolution range estimated using the FSC = 0.143 criterion. Cryo-EM density visualization was done in UCSF Chimera[70] and UCSF ChimeraX[71].

### Model building
The hTRPV6Mg structural model was built in Coot[72], using the previously published cryo-EM structure of hTRPV6 in the closed state (7S8B)[40] as a guide. The resulting model was real space refined in Phenix[73] and visualized in UCSF Chimera, UCSF ChimeraX, and Pymol[74]. The pore radius was calculated using HOLE[75].

### Electrophysiology
Patch-clamp recordings with the full-length hTRPV6 were performed as reported previously with a few modifications[12]. HEK 293T cells grown in 35-mm dishes to ~60% confluence were transiently transfected with 0.8 µg/dish of human TRPV6 cDNA in the pEG BacMam expression vector[37,40] using the Lipofectamine 2000 transfection reagent (Thermo Fisher Scientific). Patch-clamp current recordings were conducted 18–22 h after transfection from HEK 293T cells displaying eGFP fluorescence. Whole-cell currents were recorded using an EPC10 patch-clamp amplifier and PatchMaster software (Version V2x92, Harvard Bioscience). Voltages were corrected for the liquid junction potential of 10 mV. Holding potential was 0 mV. Currents were elicited by voltage ramps from −100 mV to +100 mV over 200 ms applied every 2 s. Inward or outward current amplitudes were measured at −80 or +80 mV and normalized to the cell size as pA/pF. The capacitance was measured using the automated capacitance cancellation function in EPC10. The step protocol consisted of a 100-ms pre-pulse step at −100 mV, 100-ms steps in the range of −160 to +100 mV, followed by a 100-ms step at −100 mV. The standard extracellular solution contained 140 mM NaCl, 2.8 mM KCl, 10 mM HEPES, and

## Table 1 | Cryo-EM data collection, refinement and validation statistics

| Structure | hTRPV6$_{Mg}$ |
|---|---|
| Preparation | cNW11/soybean lipids |
| EMDB accession code | EMD-49646 |
| PDB accession code | 9NQ9 |
| Data collection and processing | |
| Magnification | 105,000x |
| Voltage (kV) | 300 |
| Electron exposure (e⁻Å⁻²) | 60 |
| Defocus range (μm) | −0.8 to −2.0 |
| Reported pixel size (Å) | 0.829 |
| Exposures (no.) | 5706 |
| Processing software | |
| Motion correction | RELION v4.0 |
| CTF estimation | Gctf v1.06 |
| Platform software for particle picking | RELION v4.0 |
| Software for 2D/3D class. & Refinements | RELION v4.0 |
| Symmetry imposed | C4 |
| Initial particle images (no.) | 5,057,951 |
| Final particle images (no.) | 31,296 |
| Map resolution (Å) FSC 0.05 | 2.97 |
| Refinement | |
| Initial models used (PDB code) | 7S8B |
| Model resolution (Å) | 2.97 |
| FSC threshold | 0.143 |
| Map sharpening $B$ factor (Å²) | −65.9 |
| Model composition | |
| Non-hydrogen atoms | 19,281 |
| Protein residues | 2320 |
| Ligands | 29 |
| Water | 0 |
| $B$ factors (Å²) | |
| Protein | 53.43 |
| Ligands | 38.11 |
| Water | n.a. |
| R.m.s. deviations | |
| Bond lengths (Å) | 0.01 |
| Bond angles (°) | 1.526 |
| Validation | |
| MolProbity score | 1.40 |
| Clash score, all atoms | 6.90 |
| Poor rotamers (%) | 0.80 |
| Ramachandran plot | |
| Favored (%) | 97.91 |
| Allowed (%) | 2.09 |
| Disallowed (%) | 0 |

10 mM EDTA. The intracellular solution contained 120 mM Cs-glutamate, 8 mM NaCl, 10 mM HEPES, 10 mM EGTA, and 0, 0.5, 2 or 10 mM MgCl$_2$, as indicated in the figure legends. Solutions were adjusted to pH 7.2 using an FE20 pH meter (Mettler Toledo) and to 290 mOsm using a Vapro 5520 osmometer (Wescor Inc).The same ramp and step protocols were used in the inside-out patch-clamp recordings. The extracellular solution contained 140 mM NaCl, 2.8 mM KCl, 10 mM HEPES, and 10 mM EDTA. The intracellular solution comprised 140 mM NaCl, 2.8 mM KCl, 10 mM HEPES, and 10 mM EGTA, with or without

2 mM MgCl$_2$. Voltages were not corrected for the liquid junction potential.

## Identification of magnesium binding sites

Typically, Mg$^{2+}$ binding site includes two or more closely spaced acidic aspartate or glutamate residues[76]. The potential Mg$^{2+}$ binding sites in hTRPV6$_{Mg}$ and hTRPV6$_{Open}$ (7S8B) in the presence and absence of Mg$^{2+}$ were identified by assuming simultaneous direct (not water mediated) binding of Mg$^{2+}$ ion to carboxylic groups of at least two acidic residues (Supplementary Fig. 4). The criterion for Mg$^{2+}$-site determination was the requirement for the distance between the Cα atoms of any aspartate (D) or glutamate (E) residues to be less than 1.1–1.3 nm, depending on the residue combination: DD, DE, or EE. This distance includes the sum of distances from the carbon atoms of the carboxyl groups (Cγ) to the Cα atoms of D (0.25 nm) or E (0.35 nm) residues, as well as the double distance between Cγ and directly bound Mg$^{2+}$ (2 × 0.3 nm).

## MD modeling and calculations of the free energy of cation binding

Three model systems were constructed for molecular dynamics (MD) simulations with Mg$^{2+}$/Ca$^{2+}$ cations binding to the acetate ion ("acetate site", AS), to the D489-D580 site in TRPV6$_{Mg}$ ("protein site", PS) and to the same site in magnesium free TRPV6$_{Eco}$ ("apo protein site", PS-apo, 7S8C[40]). The AS systems were composed of the acetate ion, cation (Mg$^{2+}$ or Ca$^{2+}$) and chloride ion (Cl$^-$) added for electroneutrality (Supplementary Fig. 8a). In AS, the ions were placed into the box of $5 \times 6 \times 5$ nm³ filled with 4460 water molecules. The carbon atoms of the acetate were restrained with the force constant of $10^3$ kJ mol$^{-1}$nm$^{-1}$ to keep its orientation along the Y-axis. To enforce the sampling rate, the cation position was constrained within a cylinder with the radius $R_{cyl}$ of 0.7 nm oriented along the Y-axis. The carboxyl group of the acetate was located on one side of the cylinder to provide the cation with an opportunity to approach from any direction. The cylindrical constraints were provided by the flat-bottom potential with the force constant, $K_{res} = 10^5$ kJ mol$^{-1}$nm$^{-2}$. Cl$^-$ ions were restrained outside the cylinder, with the force constant of $10^3$ kJ mol$^{-1}$nm$^{-1}$ along every axis. The PS and PS-apo models were constructed in a similar way. Namely, the orientation of the bundle of S5 (residues 477–501) and S6 (residues 568–592) helices was chosen to provide the cation with sufficient space to approach the D489-D580 site along the Y-axis. The backbone carbon and nitrogen atoms of S5 and S6 were restrained with the force constant of $10^3$ kJ mol$^{-1}$nm$^{-1}$ along every axis. The cation position was constrained within the cylinder, which had the radius $R_{cyl}$ of 0.9 nm, oriented along the Y-axis, with $K_{res} = 10^5$ kJ mol$^{-1}$nm$^{-2}$. Cl$^-$ restrained with the force constant of $10^3$ kJ mol$^{-1}$nm$^{-1}$ along every axis. The S5-S6 bundle, cation and Cl$^-$ were placed in the $5 \times 8 \times 5$ nm³ box filled with 5550 water molecules (Supplementary Fig. 6a).

Molecular modeling of divalent cation binding to carboxylate or phosphate groups has been proven to be a challenging task[77–81]. Commonly used additive force fields (FF) for MD simulations of biomolecular systems, such as CHARMM[82] and AMBER[83], tend to overestimate the energy of divalent cations interactions with protein charged groups, because of inappropriate treatment of charge shielding induced by the solvent polarization effects. To address this issue, we employed an electronic continuum correction (ECC) approach with "scaled" FF, where atomic charges are proportionally reduced to enhance the agreement with the experimental data[48]. This method previously showed good agreement between the affinity of Mg$^{2+}$ and Ca$^{2+}$ to acetate ion measured experimentally and calculated using molecular modeling methods[80]. Initially, we employed the ECC scaling parameters for acetate, Mg$^{2+}$ and Ca$^{2+}$ proposed earlier[80], corresponding to 0.8 charge scaling factor (SF) for the SPC/E water model, where they were named as "acetate_sc3", "MG_s3", and "CA_3s". However, according to quantum chemical calculations, Ca$^{2+}$ being coordinated by oxygens of carboxyls and/or water molecules, preserves a

larger charge as compared with $Mg^{2+}$. This indicates the necessity to use a larger SF value for $Ca^{2+}$. Our test simulations showed that the increased SF = 0.85 for $Ca^{2+}$-acetate binding matches the experimental data better (see below). Therefore, for PS and PS-apo systems, we used the parameterization "MG_s3" with SF = 0.8 for $Mg^{2+}$ binding and modified parameterization with SF = 0.85 for $Ca^{2+}$. For the protein helices, we employed the Amber-99sd-ildn FF[83], with all charges scaled by the SF of 0.8 for $Mg^{2+}$ or 0.85 for $Ca^{2+}$. The chlorine ion charge was modified to ensure the electroneutrality within the systems. The SPC/E water model[84] was used in all simulations to maintain consistency with the previously published FF[80].

The standard binding free energy ($\Delta G^0$) was calculated using the well-tempered metadynamics approach (WTMetaD)[49]. The collective variables (CVs) for AS were the distance from the cation to the carbon atom of the carboxyl group of acetate (L) and the number of oxygen atoms of water molecules inside the first coordination shell of $Mg^{2+}$ or $Ca^{2+}$ ($CN_W$). In PS and PS-apo, we used three CVs, including the distance from the cation to Cγ of D489 ($L_1$) and D580 ($L_2$) and $CN_W$. The metadynamics simulations were performed using GROMACS 2024.3 software package[85] patched with PLUMED 2.9.2 library[86]. For AS, we performed WTMetaD simulations spanning 3 µs. To enhance sampling efficiency in the more complex PS and PS-apo systems, we employed a "multiple walkers" WTMetaD approach[50], simulating four parallel walkers for 2 µs each, resulting in a total simulation time of 8 µs. Simulations were carried out with an integration time of 2 fs. Hydrogen bond lengths were constrained by the LINCS algorithm[87], with imposed 3D periodic boundary conditions, constant temperature (310 K) and pressure (1 bar). The cutoff distance of 1.2 nm was used for evaluation of nonbonded interactions, and the particle-mesh Ewald method[88] was employed for treatment of long-range electrostatics. All systems were first equilibrated in several stages: $5 \times 10^4$ steps of steepest descent minimization, followed by heating from 5 to 310 K during a 200-ps MD run, and 10 ns of MD run with fixed positions of the acetate/protein and cation. Then, WTMetaD-production runs were carried out with the bias factor of 5, bias height of 0.3 $k_B$T, and gaussians width of 0.05 nm for the distances (L, $L_1$, $L_2$) and 0.1 for $CN_W$. $CN_W$ was defined using the switching function $CN_W = (1 - (r_i/r_0)^6)/(1 - (r_i/r_0)^{12})$, where $r_i$ was the distance between the cation and oxygen atom of the i-th water molecule. The values of $r_0 = 0.275$ nm for $Mg^{2+}$ and $r_0 = 0.310$ or 0.3067 nm for $Ca^{2+}$ with SF of 0.8 or 0.85 were chosen to obtain the $CN_W$ value of 6 and 7 in bulk water, respectively. All WTMetaD trajectories are listed in Supplementary Table 1.

### Calculations of the minimum free energy path and the standard binding free energy

The result of the WTMetaD simulations was the distribution of two (2D for AS) or three (3D for PS and PS-apo) dimensional free energy surfaces (FES) along CVs: L, $CN_W$ and $L_1$, $L_2$, $CN_W$, accordingly. To determine the energetically preferable binding pathway of a cation to the D489-D580 site, the projection of 3D FES of the cation-protein system in the 2D plane of two CVs $L_1$ and $L_2$ (Fig. 7a, b, Supplementary Fig. 7f, j) was calculated using the reweighting procedure[89] of the CVs distributions over the WTMetaD trajectories. The minimum free energy path (MFEP) along $FES_{2D}$ ($L_1$, $L_2$) (black lines in Fig. 7a, b) was calculated using the string method[90,91] implemented in python script (https://github.com/jvermaas/zerotemperaturestring). The standard binding free energies of cations at zero ionic strength ($\Delta G^0$) were derived from 1D FES profiles (PMFs), also calculated through reweighting (Supplementary Figs. 6h, i, 7i, m, 8e). The beginning fragments of AS, PS and PS-apo trajectories (1, 2 and 2 µs, respectively) were omitted to obtain equilibrated CVs distributions.

$\Delta G^0$ was calculated using a previously proposed approach[92]. Specifically, two regions of PMFs were considered: "bound" (L < 0.4 nm) – corresponding to the cation that resided in the site and "unbound" (2 < L < 2.5 nm) for the cation in bulk water. $\Delta G^0$ (Eq. (1)) was calculated

as the sum of the following terms: $\Delta G_{PMF}$ (Eq. (2)) calculated from the PMF profile; $\Delta G_V$ (Eq. (3)) that accounts for the ratio of the sampled unbound volume to the standard-state volume ($V^0$), incorporating the effect of the cylindrical restraint applied to the cation; and $\Delta G_I$ (Eq. (4)) that converts binding energy to standard conditions at zero ionic strength using the limiting Debye-Hückel equation[93,94]. In the equations below, $S_U$ is the cross section of the cation constraining cylinder with radius $R_{cyl}$ provided by flat-bottom potential $U_{res}(r)$ with force constant $K_{res}$, RT = 2.577 kJ/mol, $V^0 = 1.661$ $nm^3$, $l_u = 0.5$ nm is the length of the unbound region of the PMF. $\gamma_i$ represents activity coefficients for cation-site complex and cation and site separately with ionic charge $z_i$, A = 0.519 $M^{-1/2}$ in water solution at 310 K[94], I is the ionic strength presented in Supplementary Table 1.

$$\Delta G^0 = \Delta G_{PMF} + \Delta G_V + \Delta G_I \tag{1}$$

$$\Delta G_{PMF} = RT \ln \left( \frac{\int_{unbound} e^{-\frac{PMF(x_1)}{RT}} dx_1}{\int_{bound} e^{-\frac{PMF(x_1)}{RT}} dx_1} \right) \tag{2}$$

$$\Delta G_V = -RT \ln \left( \frac{l_u S_u}{V^0} \right) \tag{3}$$

where

$$S_u = \int_0^{+\infty} 2\pi r e^{-U_{res}(r)/RT} dr = \pi R_{cyl}^2 + 2\pi \left( \sqrt{\frac{\pi RT}{2K_{res}}} R_{cyl} + \frac{RT}{K_{res}} \right)$$

$$\Delta G_I = RT \ln \left( \frac{\gamma_{cation} \gamma_{site}}{\gamma_{cation-site}} \right) \tag{4}$$

where

$$\gamma_i = -A z_i^2 \sqrt{I}$$

$\Delta G^0$ values and their errors were estimated via the block analysis approach. The WTMetaD trajectories were divided into an increasing number of blocks, ranging from 3 to 100 for AS and to 25 for PS and PS-apo. For each block size the mean $\Delta G^0$ and a standard error of the mean (SEM) was calculated, and the plateau reached by these values was used as a criterion of convergence calculation (Supplementary Figs. 6b, c, 7d, e, 8f, g). $\Delta G^0$ and SEM values for the maximum block size were taken as the calculation results presented in the main text and Supplementary Table 1. Structures were visualized in Pymol[95].

### Quantum chemical calculations

The following procedure was used to build the initial geometry of the acetate ion complexes with $Mg^{2+}$ and $Ca^{2+}$ in an aqueous solution. The cation was first placed in a water box, and one of its neighboring water molecules was replaced by an acetate. Then all the waters were removed from the system, except for those that were inside the first two solvation shells of the cation. The remaining systems contained $Mg^{2+}$ and $Ca^{2+}$ and the neighboring acetate ion, buried in a cluster of 31 and 22 waters, respectively. Complexes of two acetate ions with both cations were built in a similar way with 17 and 20 water molecules. Geometry optimization for the models was performed without restraints using the Hartree-Fock method with the 6−31 G(d) basis in GAUSSIAN09, Revision A.01 software[96].

### Validation of simulation protocols

Before simulations of the PS and PS-apo systems, we performed the validation of FF parameters and calculations protocols using well-studied cation – acetate ion systems (AS system, Supplementary

Fig. 8). Firstly, we used the previously proposed parameters[80] with SF = 0.8. For magnesium-acetate, we obtained $\Delta G^0 = -3.7 \pm 0.1$ kJ/mol, which is in good agreement with the experimental values ranging from $-7.3$ to $-3.4$ kJ/mol[80,97–100] and with the previous calculation of $-5.3 \pm 0.5$ kJ/mol[80]. For $Ca^{2+}$, the same SF yielded $\Delta G^0 = 0.0 \pm 0.4$ kJ/mol, which is outside the experimental range from $-6.8$ to $-2.5$ kJ/mol. SF modification to 0.85 showed good agreement with the experiment ($\Delta G^0 = -3.0 \pm 0.2$ kJ/mol), reflecting a larger effective charge saving on $Ca^{2+}$ compared to $Mg^{2+}$ in aqueous solution, which is supported by quantum chemical calculations of partial charges in the systems containing water and/or carboxyl oxygens coordinating both cations.

The low-energy conformations of acetate ion complexes with $Mg^{2+}$ and $Ca^{2+}$ in MD simulations were compared with conformations obtained independently using quantum chemical calculations in water clusters. In both cases, the structure of the first coordination sphere of cations and cation-acetate oxygen distances (0.21 nm for $Mg^{2+}$, 0.23 nm for $Ca^{2+}$), show good agreement between MD and ab initio quantum chemical calculations. This additionally confirms the correctness of the ECC parameters used for cations binding modeling.

To ensure the reliability of the PS and PS-apo systems WTMetaD-simulations results, we additionally examined conformational sampling of the D489 and D580 side-chain dihedral angles, as these degrees of freedom have the capacity to influence the properties of PS-cation binding while remaining unbiased. To characterize the available conformational space in the PS system, we performed an unbiased MD simulation without any cation in the site ("MD-free", 5 μs). The distributions of the $\chi_1$ (N-Cα-Cβ-Cγ) and $\chi_2$ (Cα-Cβ-Cγ-O$_{D1}$) dihedral angles for both residues D489 and D580 are plotted in Supplementary Fig. 9a–c. In the MD-free simulation, multiple metastable states were observed (gray points). For D489, the angle values were $\chi_1 \approx 50°$ and $\chi_2 \approx -100°/+100°$, while D580 adopted $\chi_1 \approx -60°/180°$ and $\chi_2 \approx -100°/0°/+100°$. To verify that the dihedrals sampling in the WTMetaD simulations were complete, a reweighting procedure was performed to project FES onto the $\chi_1$-$\chi_2$ plane. The results for the PS and PS-apo systems are shown in Supplementary Fig. 9d–k. All dihedral angles available in the MD-free case were found to be sampled during WTMetaD in all trajectories. Additionally, we plotted the $\chi_1$-$\chi_2$ for the states, when the cations are bonded to the D489-D580 site (red dots in the same panels). Although statistics in this case are quite poor, it can be seen that in the bonded states WTMetaD also sampled nearly the same conformational space as in the MD-free trajectory. Thus, we can conclude that the dihedrals were properly sampled in WTMetaDs.

### Reporting summary

Further information on research design is available in the Nature Portfolio Reporting Summary linked to this article.

## Data availability

The cryo-EM density map of hTRPV6 in complex with $Mg^{2+}$ was deposited to the Electron Microscopy Data Bank (EMDB) under the accession code EMD-49646. The atomic coordinates have been deposited to the Protein Data Bank (PDB) under the accession code 9NQ9. The coordinates of the protein and ligand obtained in MD simulations are available via the Github platform under the link https://github.com/Gressy2113/TRPV6-MG.git and on Zenodo repository with the record number 17233354[101]. All other data is available from the corresponding authors upon request. Source data are provided with this paper.

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

## Acknowledgements

We thank Joanna Zaisserer and Anna Erbacher (LMU Munich) for their excellent technical assistance, Kirill Nadezhdin for help in preparing cryo-EM samples and Irina A. Talyzina for her excellent assistance in the preparation of the revised manuscript in response to reviewers' comments. Access to computational facilities of HPC facilities at NRU HSE, the Supercomputer Center "Polytechnical" at the St. Petersburg Polytechnic University and IACP FEB RAS Shared Resource Center "Far Eastern Computing Resource" equipment (https://cc.dvo.ru) is gratefully appreciated. As a Walter Benjamin Fellow, A.N. was funded by the Deutsche Forschungsgemeinschaft (DFG, German Research Foundation) – 464295817. I.I.V., Y.A.T., and R.G.E. were supported by the RSF (23-14-00313). T.G. and V.C. were supported by DFG TRR 152 (P15) and GRK 2338 RTG (P10). A.I.S. was supported by the Human Frontier Science Program (HFSP) Award and the NIH (AR078814, CA206573, NS083660, NS107253).

## Author contributions

A.N. carried out protein expression, protein purification, cryo-EM sample preparation and cryo-EM data processing. A.S., T.G. and V.C. performed and analyzed electrophysiological experiments. I.I.V, Y.A.T., and R.G.E. performed MD simulations and MD analysis. A.N. and A.S. made constructs. A.N. and A.I.S. analyzed structural data and built molecular models. A.N., A.S., V.C., I.I.V., and A.I.S. wrote the manuscript, which was then edited by all authors.

## Competing interests

The authors declare no competing interests.
