## [Transparent Peer Review file · Nature Communications]

The locking mechanism of human TRPV6 inhibition by intracellular magnesium

Corresponding Author: Dr Alexander Sobolevsky

Version 0:

Reviewer comments:

Reviewer #1

(Remarks to the Author)

In the manuscript by Neuberger and colleagues, the authors examine the mechanism by which magnesium ions regulate the human TRPV6 channel. TRPV6 plays an important role in mediating Ca²⁺ uptake across various tissues, including in epithelial cells of the intestinal tract. The channel is also overexpressed in several malignancies. Mutations and abnormal expression of TRPV6 are linked to various human diseases. While the structure of the open state of TRPV6 has been previously resolved, the structure of the closed state has remained undetermined, constituting a significant barrier to the development of TRPV6 inhibitors that could be clinically utilized.

Here, the authors resolve the structure of hTRPV6 in the presence of Mg²⁺ ions using CryoEM. Several putative Mg²⁺ binding sites were identified. Mutational analysis combined with electrophysiology experiments was used to confirm TRPV6 D580 as an important residue involved in binding Mg²⁺. Application of Mg²⁺ causes block of TRPV6 inward and outward currents. Mutation of D580 to asparagine, arginine, or alanine produced differential changes in the extent of Mg block. The D580R and D580A also affected the kinetics of current relaxation of inward currents when the voltage was stepped to -100 mV. These experiments strongly support the identification of D580 as a residue important to the binding pocket for Mg²⁺, which also involves D489. Unfortunately, mutation of D489 to alanine or asparagine disrupted the channel, preventing study of the impact of D489 on Mg²⁺ control of the TRPV6 channel. Elegant modeling further supports the identification of D489 and D580 as the Mg²⁺ binding site in TRPV6.

While Mg²⁺ has been shown to block other TRP channels, the Mg²⁺ binding site in TRPV6 is unique. Importantly, the resolved structure of the inactivated TRPV6 channel will be extremely helpful to the rational design of TRPV6 therapeutics.

Overall, the work reported by Neuberger and colleagues is highly significant to the TRPV6 field, advancing understanding of the mechanistic control of the TRPV6 channel. All of the obtained data support the conclusions of the paper. The data analysis conducted was rigorous, and appropriate statistical analysis was performed throughout the manuscript. The methodology employed in the study was of extremely high quality, and the methods section provided sufficient detail for the work to be reproduced. Overall, the paper reports an important structural model of the inactivated state of TRPV6, which will be highly valued by the TRPV field.

I have only minor suggestions/comments for the authors.

1. In Figure 5e, the kinetics of current relaxation of inward currents when the voltage was changed from +40 to -100 mV are different between the first and second step. In the second step, the relaxation rate was nearly instantaneous. Was this typically observed for this mutant? This might be additional evidence that the mutation destabilizes the inactivated state of the channel.
2. In the discussion section, it would be helpful if the authors could speak about any potential physiological relevance of Mg²⁺ inhibition of the channel.

Reviewer #2

(Remarks to the Author)

The manuscript entitled "The locking mechanism of human TRPV6 inhibition by intracellular magnesium" by Arthur Neuberger et al. presents a comprehensive study of the structure and function of the TRPV6 channel. As a selective calcium channel, TRPV6 plays a central role in calcium uptake and homeostatic regulation, and its dysregulation or overactivation has been linked to several diseases. Previous studies have shown that intracellular magnesium ions (Mg^{2+}) inhibit TRPV6 currents, a critical aspect of its physiological regulation. However, the precise mechanism of inhibition remained unclear. In this work, the authors employ an integrative approach that combines single-particle cryo-electron microscopy (cryo-EM), electrophysiology, molecular dynamics simulations, and biochemical assays to unravel the molecular basis of Mg^{2+} -mediated TRPV6 inhibition. The key finding of the study is the identification of a novel Mg^{2+} binding site near the intracellular pore entrance of each TRPV6 subunit. Interestingly, Mg^{2+} does not directly occlude the channel, but instead stabilises a closed conformation, thereby inhibiting calcium permeation. The authors further validate this model using electrophysiological recordings and molecular dynamics simulations, strengthening its mechanistic plausibility. This work significantly advances our understanding of TRPV6 regulation and provides a structural basis for the rational design of TRPV6-targeting therapeutics. In addition, the findings may shed light on the pathogenesis of TRPV6-related disorders and provide insights for optimising treatment strategies.

Minor suggestions:

The authors impose C4 symmetry during 3D refinement, implying equivalent Mg^{2+} binding across all subunits. However, given the potential resolution limitations, it would be valuable to:

Present a C1 reconstruction (if feasible) to support the proposed binding stoichiometry.

To strengthen the structural interpretation, the authors could provide an EM density around the Mg^{2+} binding site (e.g. in Figure 3B).

Otherwise, the manuscript is well presented, carefully analyzed, and supported by robust experimental and computational data. The figures are clear and effectively illustrate the results. With the minor revisions addressed, this paper is very suitable for publication in Nature Communications.

Reviewer #3

(Remarks to the Author)

Neuberger et al. have presented a significant and timely contribution to the field by investigating the inhibition of the calcium-selective TRPV6 channel, a protein of considerable interest in the context of cancer biology. The authors report the identification of a previously uncharacterized intracellular Mg^{2+} binding site and propose a novel mechanism by which Mg^{2+} modulates TRPV6 activity. This represents a potentially important advancement, given the role of TRPV6 in cellular calcium homeostasis and its overexpression in several cancer types.

The manuscript is strengthened by a combination of cryo-EM structural data and electrophysiological recordings, which together provide initial evidence supporting the presence and functional relevance of this intracellular Mg^{2+} site. However, despite the novelty and potential significance of these findings, there remain substantial gaps in experimental validation that limit the strength of the conclusions. In particular, the functional data do not yet fully establish a direct causal relationship between the observed density in the structure and Mg^{2+} -mediated channel inhibition under physiological conditions.

To solidify the claims and enhance the mechanistic insight into this mode of inhibition, I strongly recommend the authors address the following key experimental gaps:

1) The claimed Mg^{2+} specificity of the D489 and D580 site is inferred primarily from computational comparisons with Ca^{2+} , but no experimental assessment of selectivity against other divalent cations (Zn^{2+} , Mn^{2+}) is provided. Functional testing with a panel of divalent ions would strengthen the claim of Mg^{2+} specificity;

2) the loss-of-function phenotype of the D489A and D580A mutants requires additional validation—either via surface expression assays to confirm proper trafficking, or by rescue with conservative substitutions (I would suggest D489E) to test coordination requirements;

3) The use of whole-cell patch-clamp leaves uncertainty regarding actual intracellular Mg^{2+} concentrations; I would highly recommend additional recordings in inside-out configuration to more precisely control the intracellular environment;

4) Mg^{2+} ion was assigned without resolving it directly. The assignment of the cryo-EM density as Mg^{2+} would benefit from independent validation, such as anomalous scattering with Mn^{2+} (X ray) or structural analysis of Mg^{2+} -site mutants (cryo-EM of D580R and) to confirm disappearance of the density;

5) The D580R mutant exhibits significant effects on TRPV6 function in Mg^{2+} -free conditions with notably reduced currents. Given this apparent functional impairment caused by the mutation, characterization of Mg^{2+} -dependence may not yield meaningful insights since channel properties are likely altered. It might be more relevant to focus on mutants that preserve channel function and current amplitude in the absence of Mg^{2+} .

6) To establish the link between Mg^{2+} -mediated inhibition and the proposed binding residues (D489, D580), functional characterization of double mutants could be beneficial. Ideally, if these residues lose their affinity for Mg^{2+} , the Mg^{2+} -induced locking of the channel's closed state should be relieved. This would further strengthen the proposed mechanism underlying TRPV6 inhibition.

7) Was a leak control (I would suggest to use La^{3+}) applied at the end of the electrophysiology experiments? This approach would allow for a precise assessment of the channel's properties, particularly since the measurements indicate relatively high outward currents.

Most importantly, please know that my suggestions, although numerous, are made with the greatest respect and genuine appreciation for the work presented. My intention is not to criticize, but rather to help strengthen the key findings and ensure

that the most important conclusions are supported as convincingly as possible. I truly hope the authors do not feel discouraged or offended—on the contrary, I believe this study has strong potential, and my comments are meant in the spirit of constructive and collegial dialogue.

Reviewer #4

(Remarks to the Author)

Reviewer #5

(Remarks to the Author)

The manuscript by Neuberger et al. with the title "The locking mechanism of human TRPV6 inhibition by intracellular magnesium" describes a combined experimental-computational study that aims to identify the Mg²⁺ binding site and uncover the mechanism of inhibition by Mg²⁺. The manuscript shows the reduction of current in the presence of Mg²⁺ and goes on to determine the Mg²⁺ bound structure of TRPV6. The channel was determined in the closed conformation. Additional density was observed in the selectivity filter and in a new binding site in the interface between S5 and S6. This putative Mg²⁺ regulatory site was found to coordinate the Mg²⁺ by D439 and D580 in the closed state, while rotation of the S6 would disrupt the regulatory binding site in the open state.

The authors further investigated the proposed regulatory site by site-directed mutagenesis, electrophysiological recordings and MD simulations. The manuscript is well written and the figures are clear.

Despite this merit, I do have doubts about the interpretation of the main results.

Generally, earlier electrophysiological results should more extensively be taken into considerations in the current manuscript. The paper of Voets et al from 2003 (citation #8) interpretes the electrophysiological recordings differently and reports that Mg²⁺ would bind in the premeation path. The this paper also showed that the channel can be equally blocked by intracellular or extracellular Mg²⁺. How would it be possible to imagine that intracellular and extracellular Mg²⁺ could lead to the same interaction kinetics, if the Mg²⁺ binding site would be in the proposed place?

In the selectivity filter, a calcium ion was modelled by the authors. The experimental procedure leading to channel structure determination was Ca²⁺ free. While free of divalent ions throughout most of the purification procedures, 2 mM Mg²⁺ were added at the end and were allowed to equilibrate for 1h before freezing to the grid. How can the authors be sure that the density in the selectivity filter identified as Ca²⁺ is in reality not a Mg²⁺? Please find an approach to verify the identity of the ion in the selectivity filter of the sample, if not identical to Mg²⁺?

The paper by Jean et al from 2002, (citation #42) showed that residues lining the selectivity filter are the locus responsible for Ca²⁺ and Mg²⁺ binding and premeation block. The authors of the current manuscript find residual density in the selectivity filter. Could this density be the site identified in the 2002 paper? To assess (to rule out or to confirm) whether the selectivity filter is the site of Mg²⁺ block and of Ca²⁺ binding, key residues in close proximity to the identified density should be mutated and investigated by electrophysiological recordings.

It remains unclear how figure 1d would be explainable by the proposed mechanism? The proposed Mg²⁺ binding site is only formed in the closed state. In the presence of Mg²⁺, Figure 1d shows increasing inward current which slowly grows until ~100s. At this point, ~80% of all channels are open (by comparing to the current at 0 Mg²⁺). In the open geometry, Mg²⁺ binding should not occur, as the binding site was destroyed by the rotation of the S6. Once the voltage is shifted to +40 mV, all channels close immediately. One would expect a much slower (possibly slower than the opening rate) channel closing rate, as the S6 has first to rotate to form the proposed Mg²⁺ binding site, and only then could Mg²⁺ bind to the proposed site.

From the perspective of a physiological role, the competition with Ca²⁺ is missing. I would be very helpful to carry out experiments with increasing extracellular Ca²⁺ concentrations and quantify currents of WT and mutants (including a least one mutant in the selectivity filter).

Figure 5. The D580 mutants are designed to affect Mg²⁺ binding by removing one of the key aspartate residues interacting with Mg²⁺. It is puzzling, how an inward rectification by a Mg²⁺ block at +40 is still possible in these mutants, as it should not bind Mg²⁺.

Figure 5e shows a much faster opening kinetics for D580R as compared to all other tested D580 variants, also compared to WT. Something important must be different. Could the authors investigate the reason, or at least speculate on the reason of this observation?

Figure 6. The D580R mutant exchanges the negatively charged aspartate with the positively charged arginine. In the closed channel state, the arginine side chain will be rotated towards the position, where the Mg²⁺ could be coordinating D498 and D580R. How is it explainable that a Mg²⁺ (at 10 mM) bound to D498 at the putative bind site could stabilize the closed conformation instead of further destabilizing it? In the presence of Mg²⁺, the positive charge at D580R would be expected to be pushed away (from close to open) enhancing the open state of the channel, possibly also at positive voltages. This seems a contradiction.

Free Energy: acetate reference system: The authors write on line 201 to 205: "The resulting FES and ΔG_0 values of cation-acetate binding, -3.2 ± 0.5 kJ/mol for Mg²⁺ and $+0.5 \pm 0.3$ kJ/mol for Ca²⁺, were in reasonable agreement with the experimental data ranging from -7.3 to -3.4 kJ/mol for Mg²⁺ and -6.8 to -2.5 kJ/mol for Ca²⁺ ions, and with the previously reported simulation results (-7.0 to -5.2 kJ/mol for Mg²⁺ and -5.0 to -0.8 kJ/mol for Ca²⁺)". The authors thus report for the acetate binding simulations a value $+0.5$ kJ/mol for Ca²⁺, which would mean repulsive instead of attraction. Previous computational data reported -5.0 to -0.8 kJ/mol, while experimental data showed -6.8 to -2.5 kJ/mol. This is far from "a reasonable agreement", indicating that the computational approach must suffer from a fundamental shortcoming, suggesting that the data cannot be seen as reliable.

On lines 226 to 230, the authors write: "We observed a significant difference between ΔG_0 for binding of Mg²⁺ (-11 ± 1 kJ/mol) and Ca²⁺ (-3.0 ± 0.2 kJ/mol) to the D489-D580 site, which corresponds to about 20 times higher preference in binding of Mg²⁺ compared to Ca²⁺. Thus, the D489-D580 site appears to be a magnesium-binding site with high selectivity to Mg²⁺ compared to Ca²⁺". The discrepancies observed for the free energy in the aspartate example questions the validity of this observation and needs further verification. Also, the stated difference is contradicted by the data shown in Figure 7 and S6, which show no significant differences beyond higher energy barriers for Mg²⁺, opposing binding of Mg²⁺. What is the rationale for this statement?

Free Energy, PS system: The interpretation of the data could be complicated, if e.g. the sidechain of D498 or D580 would rotate while the divalent ion position remains unchanged. To be sure that this is not the case and phase space remains connected, please report on ch1 of the two aspartate residues.

Free Energy method: The divalent ion has been restrained to a cylinder. How was the influence of this restraint corrected for?

Amber was parametrized together with the TIP3P water model. Please cite a study confirming that Amber-99sd-ildn can be combined with the SPC/E water model.

It is unexpected to see very different values and block size dependent trends for Mg²⁺ and Ca²⁺. A change of ~ 9 kJ/mol and changing sign as a function of block size is an alarming signal. Setting the preferred block size value would then allow for obtaining the sought results. Please carefully investigate the source of the striking and alarming difference.

In the QM calculate, only two water layers were included. It is well known that hydrogen bond mediated water structures orient water molecules way beyond two water layers. If waters are removed beyond two layers, strong perturbations have to be expected. Please correct for this shortcoming or remove the QM part.

It might be interesting to assess, if the Mg²⁺ dependency of channel permeations is conserved across the TRPV family in the same way as its D498 - D580 residue pair.

Please consistently use nm or Å throughout the manuscript instead of continuously switching units.

Version 1:

Reviewer comments:

Reviewer #1

(Remarks to the Author)

The authors have completed a very thorough response to the critiques of all the reviewers. I do not have any additional concerns.

Reviewer #2

(Remarks to the Author)

After reading the revised version, I think the authors have provided a good response, and all the issues I raised have been properly addressed. I have no further concerns. The revised version is clearly much improved, so I recommend its acceptance.

Reviewer #3

(Remarks to the Author)

The authors have addressed my previous comments thoroughly. They provided additional experimental data, clarified

limitations where necessary, and revised the manuscript accordingly. I am satisfied with their responses, and I find the final version of the manuscript suitable for publication.

Reviewer #4

(Remarks to the Author)

Reviewer #5

(Remarks to the Author)

The authors have strongly improved the manuscript, and successfully addressed almost all of my questions. Importantly, the computational data were expanded and clarified to allow for fully comprehending the data.

Minor concerns

Regarding Ca²⁺, please also modify the statement in line 114, replacing calcium with magnesium.

We are very thankful to the Reviewers for their excellent suggestions. We have made changes in the manuscript accordingly to the details outlined in our responses below.

Reviewer #1 (Remarks to the Author)

In the manuscript by Neuberger and colleagues, the authors examine the mechanism by which magnesium ions regulate the human TRPV6 channel. TRPV6 plays an important role in mediating Ca²⁺ uptake across various tissues, including in epithelial cells of the intestinal tract. The channel is also overexpressed in several malignancies. Mutations and abnormal expression of TRPV6 are linked to various human diseases. While the structure of the open state of TRPV6 has been previously resolved, the structure of the closed state has remained undetermined, constituting a significant barrier to the development of TRPV6 inhibitors that could be clinically utilized.

Here, the authors resolve the structure of hTRPV6 in the presence of Mg²⁺ ions using CryoEM. Several putative Mg²⁺ binding sites were identified. Mutational analysis combined with electrophysiology experiments was used to confirm TRPV6 D580 as an important residue involved in binding Mg²⁺. Application of Mg²⁺ causes block of TRPV6 inward and outward currents. Mutation of D580 to asparagine, arginine, or alanine produced differential changes in the extent of Mg block. The D580R and D580A also affected the kinetics of current relaxation of inward currents when the voltage was stepped to -100 mV. These experiments strongly support the identification of D580 as a residue important to the binding pocket for Mg²⁺, which also involves D489. Unfortunately, mutation of D489 to alanine or asparagine disrupted the channel, preventing study of the impact of D489 on Mg²⁺ control of the TRPV6 channel. Elegant modeling further supports the identification of D489 and D580 as the Mg²⁺ binding site in TRPV6.

While Mg²⁺ has been shown to block other TRP channels, the Mg²⁺ binding site in TRPV6 is unique. Importantly, the resolved structure of the inactivated TRPV6 channel will be extremely helpful to the rational design of TRPV6 therapeutics.

Overall, the work reported by Neuberger and colleagues is highly significant to the TRPV6 field, advancing understanding of the mechanistic control of the TRPV6 channel. All of the obtained data support the conclusions of the paper. The data analysis conducted was rigorous, and appropriate statistical analysis was performed throughout the manuscript. The methodology employed in the study was of extremely high quality, and the methods section provided sufficient detail for the work to be reproduced. Overall, the paper reports an important structural model of the inactivated state of TRPV6, which will be highly valued by the TRPV field.

We thank Reviewer #1 for kind words about our work.

I have only minor suggestions/comments for the authors.

1. In Figure 5e, the kinetics of current relaxation of inward currents when the voltage was changed from +40 to -100 mV are different between the first and second step. In the second step, the relaxation rate was nearly instantaneous. Was this typically observed for this mutant? This might be additional evidence that the mutation destabilizes the inactivated state of the channel.

Reviewer 1 is correct; the current relaxation of D580R was different from WT TRPV6. This assumption aligns with our results in Figures 6c and 6d. Please note that the amplitude of the

tail currents at -100 mV was significantly different in WT and D580R at holding potentials positive to -40 mV (Figure 6c, d).

Nevertheless, we are more cautious in interpreting the data for D580R in Figure 5e. We believe that the difference between the first and second steps most likely reflects the fact that the first switch was from 0 to -100 mV and the second one was from +60 to -100 mV (not +40 mV, we corrected this typo in Figure 5 legend). In Figure R1 below, we plotted the data from the experiment shown in Figure 5e (+2 mM Mg²⁺), using two traces for the second switch (i) from 0 to -100 mV versus (ii) +60 to -100 mV. In this case, the relaxation in the left and right segments of the measurements was nearly the same.

Figure R1. Representative tail currents measured at -100 mV for the D580R TRPV6 channel in the presence of 2 mM Mg²⁺. Currents over time are shown for the three 100 ms voltage steps: -100 to 0 and back -100 mV (black line) or -100 to +60 then -100 mV (grey line) applied from 0 mV holding potential.

2. In the discussion section, it would be helpful if the authors could speak about any potential physiological relevance of Mg²⁺ inhibition of the channel.

We thank Reviewer #1 for the suggestion. We have added a section to the Discussion (lines 300-305) that highlights the physiological relevance of the TRPV6 channel inhibition by Mg²⁺.

Reviewer #2 (Remarks to the Author)

The manuscript entitled "The locking mechanism of human TRPV6 inhibition by intracellular magnesium" by Arthur Neuberger et al. presents a comprehensive study of the structure and function of the TRPV6 channel. As a selective calcium channel, TRPV6 plays a central role in calcium uptake and homeostatic regulation, and its dysregulation or overactivation has been linked to several diseases. Previous studies have shown that intracellular magnesium ions (Mg²⁺) inhibit TRPV6 currents, a critical aspect of its physiological regulation. However, the precise mechanism of inhibition remained unclear. In this work, the authors employ an integrative approach that combines single-particle cryo-electron microscopy (cryo-EM), electrophysiology, molecular dynamics simulations, and biochemical assays to unravel the molecular basis of Mg²⁺-mediated TRPV6 inhibition. The key finding of the study is the identification of a novel Mg²⁺ binding site near the intracellular pore entrance of each TRPV6 subunit. Interestingly, Mg²⁺ does not directly occlude the channel, but instead stabilizes a closed conformation, thereby inhibiting calcium permeation. The authors further validate this model using electrophysiological recordings and molecular dynamics simulations, strengthening its mechanistic plausibility. This work significantly advances our understanding of TRPV6

regulation and provides a structural basis for the rational design of TRPV6-targeting therapeutics. In addition, the findings may shed light on the pathogenesis of TRPV6-related disorders and provide insights for optimising treatment strategies.

We thank Reviewer #2 for the generous assessment of our work.

Minor suggestions:

The authors impose C4 symmetry during 3D refinement, implying equivalent Mg^{2+} binding across all subunits. However, given the potential resolution limitations, it would be valuable to:

Present a C1 reconstruction (if feasible) to support the proposed binding stoichiometry.

We are thankful to Reviewer #2 for the great suggestion. We have run the 3D reconstruction in C1 and observed clearly visible densities for 3 out of 4 Mg^{2+} sites (Figure S3b-f). While it is tempting to use these results to speculate about binding stoichiometry, it is possible that classification of particles with the difference of just one Mg^{2+} ion per TRPV6 subunit is not efficient (very small signal for Mg^{2+} compared to large signal for 4-fold symmetrical TRPV6 protein), and the resulting C1 map represents a mix of different stoichiometries. Technically, even one Mg^{2+} ion per TRPV6 subunit (4 Mg^{2+} ions per TRPV6 tetramer) stoichiometry is possible as part of the ensemble, especially given the weakness of Mg^{2+} density in general and not ideal overall resolution (3.35 Å in C1). Nevertheless, we have added the corresponding discussion to the text (lines 119-125).

To strengthen the structural interpretation, the authors could provide an EM density around the Mg^{2+} binding site (e.g. in Figure 3B).

We have added example densities for Mg^{2+} and surrounding protein in C4 and C1 to Figure S3b-f.

Otherwise, the manuscript is well presented, carefully analyzed, and supported by robust experimental and computational data. The figures are clear and effectively illustrate the results. With the minor revisions addressed, this paper is very suitable for publication in Nature Communications.

We appreciate the kind words of Reviewer #2 about our work.

Reviewer #3 (Remarks to the Author)

Neuberger et al. have presented a significant and timely contribution to the field by investigating the inhibition of the calcium-selective TRPV6 channel, a protein of considerable interest in the context of cancer biology. The authors report the identification of a previously uncharacterized intracellular Mg^{2+} binding site and propose a novel mechanism by which Mg^{2+} modulates TRPV6 activity. This represents a potentially important advancement, given the role of TRPV6 in cellular calcium homeostasis and its overexpression in several cancer types.

We appreciate Reviewer #3's note regarding the importance of our work.

The manuscript is strengthened by a combination of cryo-EM structural data and electrophysiological recordings, which together provide initial evidence supporting the presence and functional relevance of this intracellular Mg^{2+} site. However, despite the novelty and

potential significance of these findings, there remain substantial gaps in experimental validation that limit the strength of the conclusions. In particular, the functional data do not yet fully establish a direct causal relationship between the observed density in the structure and Mg²⁺-mediated channel inhibition under physiological conditions.

To solidify the claims and enhance the mechanistic insight into this mode of inhibition, I strongly recommend the authors address the following key experimental gaps:

1) The claimed Mg²⁺ specificity of the D489 and D580 site is inferred primarily from computational comparisons with Ca²⁺, but no experimental assessment of selectivity against other divalent cations (Zn²⁺, Mn²⁺) is provided. Functional testing with a panel of divalent ions would strengthen the claim of Mg²⁺ specificity;

We agree with Reviewer #3 that the Mg²⁺ specificity of the D489-D580 site is inferred primarily from computational comparisons. Therefore, to adjust our statements on this topic, we have removed this notion from the Discussion section. In terms of structural data, we observed that hTRPV6 displays a closed conformation upon the addition of 2 mM Mg²⁺, which should also be considered. Please note also that we do not explicitly claim in the manuscript that other divalent cations are unable to regulate hTRPV6 currents through this site. To this end, we believe that our conclusions align with the experimental evidence provided.

As suggested by Reviewer #3, we were eager to study the effects of other divalent cations on hTRPV6 currents. To this end, we first attempted to remove EGTA from intracellular solutions because this agent efficiently chelates not only Ca²⁺ but also Zn²⁺ (K_d for Zn²⁺ is ~1 nM)^{1,2}. A similar notion also applies to Mn²⁺ or for EDTA, an alternative chelator of divalent cations (<https://somapp.ucdmc.ucdavis.edu/pharmacology/bers/maxchelator/xlsconstants.htm>). Figure R2 shows whole-cell currents measured from cells expressing WT hTRPV6 using our standard intracellular solution with or without EGTA, compared to untransfected cells. We found that TRPV6 currents were not detectable upon removal of EGTA, likely because traces of Ca²⁺ in the μM range are present in unbuffered solutions³⁻⁵, which are sufficient to block Na⁺ inward currents conducted by overexpressed TRPV6⁶⁻⁸. Hence, for technical reasons, we were not able to compare the selectivity of Mg²⁺ over other divalent cations in the present study.

Figure R2. Averaged I-V relationships of whole-cell currents (mean ± SE) measured from hTRPV6-transfected and untransfected HEK293 cells in the absence or presence of 10 mM intracellular EGTA in the intracellular solution using the ramp protocol from Figure 1a. Extracellular solution contained 140 mM NaCl, 2.8 mM KCl, 10 mM HEPES, and 10 mM EDTA. The intracellular solution contained 120 mM Cs-glutamate, 8 mM NaCl, 10 mM HEPES, with or without 10 mM EGTA; n is the number of cells examined.

2) the loss-of-function phenotype of the D489A and D580A mutants requires additional

validation—either via surface expression assays to confirm proper trafficking, or by rescue with conservative substitutions (I would suggest D489E) to test coordination requirements;

Please note that D580A is not a loss-of-function mutation (Figure 5). The revised manuscript contains new data for the D580E mutation (Figure 5), showing that this TRPV6 mutant variant is similar to the WT channel, supporting the notion that the negatively charged side chain of D580 is crucial for Mg^{2+} effects. As suggested, we tested the D489E variant, which was found to be inactive (Figure 5). From the structural perspective, D489 likely plays a crucial role in stabilizing the channel gate, most likely underpinning the high sensitivity of the hTRPV6 channel to mutations at this residue. The loss-of-function mutations in TRPV6 may (or may not) alter the subcellular distribution of the mutant channels. However, we are unsure whether these changes provide mechanistic information on the role of D489 in Mg^{2+} regulation of hTRPV6 channel activity and, therefore, did not perform these experiments.

3) The use of whole-cell patch-clamp leaves uncertainty regarding actual intracellular Mg^{2+} concentrations; I would highly recommend additional recordings in inside-out configuration to more precisely control the intracellular environment;

As suggested, we conducted such experiments (new Figure S5) and found that this approach confirmed our results obtained using the whole-cell recordings.

4) Mg^{2+} ion was assigned without resolving it directly. The assignment of the cryo-EM density as Mg^{2+} would benefit from independent validation, such as anomalous scattering with Mn^{2+} (X ray) or structural analysis of Mg^{2+} -site mutants (cryo-EM of D580R and) to confirm disappearance of the density;

Unfortunately, at the current stage of cryo-EM methodology development, there are no tools like in X-ray crystallography to verify the identity of ion represented by density. We are also hesitant to solve structures of mutant channels, as the lack of putative Mg^{2+} density in them might be equally a result of loss of Mg^{2+} or altered conformation of the protein at the Mg^{2+} binding site. On the other hand, the structure solved in the absence of Mg^{2+} does not show densities in the putative Mg^{2+} sites, supporting the identified Mg^{2+} binding sites.

5) The D580R mutant exhibits significant effects on TRPV6 function in Mg^{2+} -free conditions with notably reduced currents. Given this apparent functional impairment caused by the mutation, characterization of Mg^{2+} -dependence may not yield meaningful insights since channel properties are likely altered. It might be more relevant to focus on mutants that preserve channel function and current amplitude in the absence of Mg^{2+} .

Thank you for this valuable note. We provided an additional analysis of the D580A and D580N variants to demonstrate that they also affect the response to Mg^{2+} (new Figure 5h,i). We selected D580R for more detailed analysis (i) due to its more pronounced impact on Mg^{2+} effects and (ii) because in the absence of Mg^{2+} , D580R displayed the I-V relationship analogous to the WT channel (Figure 6). Furthermore, high 10 mM Mg^{2+} concentrations could suppress D580R currents similarly to those of the WT channel (Figure 6). Overall, we believe that D580R retains key features of the channel and is well-suited for the functional assessment of Mg^{2+} effects on the channel.

6) To establish the link between Mg^{2+} -mediated inhibition and the proposed binding residues (D489, D580), functional characterization of double mutants could be beneficial. Ideally, if these residues lose their affinity for Mg^{2+} , the Mg^{2+} -induced locking of the channel's closed state

should be relieved. This would further strengthen the proposed mechanism underlying TRPV6 inhibition.

All created TRPV6 variants with mutations at D489 were found to be inactive (Figure 5). It is unclear why the additional exchange of D580 would rescue the channel activity of the double mutant. To this end, the overall rationale for performing such experiments is uncertain to us.

7) Was a leak control (I would suggest to use La^{3+}) applied at the end of the electrophysiology experiments? This approach would allow for a precise assessment of the channel's properties, particularly since the measurements indicate relatively high outward currents.

Thank you for stressing such an important issue. We systematically assessed untransfected cells to ensure that the poor patch quality and endogenous currents did not compromise our recordings (Figure 5a). Additionally, several other aspects were considered to ensure the quality of our recordings. For instance, our seals mostly exceeded $1\text{ G}\Omega$ with outward currents below 10 pA/pF immediately after break-in. At the end of the experiments, we changed the divalent-free solution to the solution containing 140 mM NaCl , 2 mM Ca^{2+} , and 1 mM Mg^{2+} , enabling us to suppress TRPV6 currents and estimate patch quality (see an example in Figure R3). We avoid using La^{3+} because this cation can lead to technical pitfalls in verifying the absence of leak currents⁹.

Figure R3. Representative currents (pA/pF) over time measured in HEK293 cells overexpressing the WT TRPV6 channel. (a) Currents at +80 and -80 mV were measured using the ramp protocol as in Figure 1a. The divalent-free solution (DVF) was exchanged for an external solution containing 2 mM Ca^{2+} , 1 mM Mg^{2+} . (b) I-V relationship at the beginning and end of measurements in (a). (c) Zoom in on outward currents in the experiment shown in (b). Extracellular solution contained: 140 mM NaCl, 2.8 mM KCl, 10 mM HEPES, and 10 mM EDTA (DVF) or 140 mM NaCl, 2.8 mM KCl, 2 mM CaCl_2 , 1 mM MgCl_2 , 10 mM HEPES-NaOH. The intracellular solution contained 120 mM Cs-glutamate, 8 mM NaCl, 10 mM HEPES, 10 mM EGTA, and 2 mM MgCl_2 .

Most importantly, please know that my suggestions, although numerous, are made with the greatest respect and genuine appreciation for the work presented. My intention is not to criticize, but rather to help strengthen the key findings and ensure that the most important conclusions are supported as convincingly as possible. I truly hope the authors do not feel discouraged or offended—on the contrary, I believe this study has strong potential, and my comments are meant in the spirit of constructive and collegial dialogue.

We truly appreciate the kind words of Reviewer #3 about our work.

Reviewer #4 (Remarks to the Author)

We appreciate the contribution of Reviewer #4 to reviewing our manuscript.

Reviewer #5 (Remarks to the Author):

The manuscript by Neuberger et al. with the title "The locking mechanism of human TRPV6 inhibition by intracellular magnesium" describes a combined experimental-computational study that aims to identify the Mg²⁺ binding site and uncover the mechanism of inhibition by Mg²⁺. The manuscript shows the reduction of current in the presence of Mg²⁺ and goes on to determine the Mg²⁺ bound structure of TRPV6. The channel was determined in the closed conformation. Additional density was observed in the selectivity filter and in a new binding site in the interface between S5 and S6. This putative Mg²⁺ regulatory site was found to coordinate the Mg²⁺ by D439 and D580 in the closed state, while rotation of the S6 would disrupt the regulatory binding site in the open state.

The authors further investigated the proposed regulatory site by site-directed mutagenesis, electrophysiological recordings and MD simulations. The manuscript is well written and the figures are clear.

Despite this merit, I do have doubts about the interpretation of the main results.

We appreciate the thorough assessment of our work by Reviewer #5. We made considerable efforts to revise our manuscript and provide a more accurate interpretation of our results.

Generally, earlier electrophysiological results should more extensively be taken into consideration in the current manuscript. The paper of Voets et al from 2003 (citation #8) interprets the electrophysiological recordings differently and reports that Mg²⁺ would bind in the permeation path. This paper also showed that the channel can be equally blocked by intracellular or extracellular Mg²⁺. How would it be possible to imagine that intracellular and extracellular Mg²⁺ could lead to the same interaction kinetics, if the Mg²⁺ binding site would be in the proposed place?

Based on available structural information about TRPV6 and the fact that the TRPV6 channel is well permeable to Mg²⁺, it is well possible that extracellular Mg²⁺ will pass the TRPV6 channel pore and, consequently, inhibit TRPV6 current via the intracellular site. Interestingly, Voets et al. also considered this option since they stated: "*The simplest model for voltage-dependent block by a permeant blocker consists of a single binding site in the channel pore that can be reached by Mg²⁺ from either side of the membrane and from which Mg²⁺ can be released to either side.*" Moreover, Voets et al. proposed that TRPV6 is regulated by Mg²⁺ through two alternative mechanisms. To this end, we do not think that the study of Voets et al. contradicts our model.

In the selectivity filter, a calcium ion was modelled by the authors. The experimental procedure leading to channel structure determination was Ca²⁺ free. While free of divalent ions throughout most of the purification procedures, 2 mM Mg²⁺ were added at the end and were allowed to

equilibrate for 1h before freezing to the grid. How can the authors be sure that the density in the selectivity filter identified as Ca²⁺ is in reality not a Mg²⁺? Please find an approach to verify the identity of the ion in the selectivity filter of the sample, if not identical to Mg²⁺?

We apologize for the confusion. In fact, the coordinate file submitted to the PDB has Mg²⁺ at the D542 site, not Ca²⁺. We absolutely agree with Reviewer #5 that Mg²⁺ is the most likely ion to occupy this site in the presence of 10 mM Mg²⁺ and absence of Ca²⁺. The reason why Ca²⁺ was originally used to model density at the D542 site (and this is why it showed up in the figures, which we forgot to modify) is because all previous structures solved in the absence of added Mg²⁺ showed a very similar density, which was modelled as Ca²⁺ based on high Ca²⁺ selectivity of TRPV6 and the results of MD simulations¹⁰. We also confirmed the binding of Ca²⁺ at this site when solving TRPV6 crystal structures and used anomalous scattering of Ca²⁺ to verify its presence^{11,12}. Unfortunately, single-particle cryo-EM has no tools alike those in crystallography to distinguish Ca²⁺ from Mg²⁺. However, in this case, given the previous functional results and conditions of our experiment, the D542 site is most certainly occupied by Mg²⁺. Accordingly, we replaced labels of Ca²⁺ to Mg²⁺ at the D542 site in all our figures.

The paper by Jean et al from 2002, (citation #42) showed that residues lining the selectivity filter are the locus responsible for Ca²⁺ and Mg²⁺ binding and premeation block. The authors of the current manuscript find residual density in the selectivity filter. Could this density be the site identified in the 2002 paper? To assess (to rule out or to confirm) whether the selectivity filter is the site of Mg²⁺ block and of Ca²⁺ binding, key residues in close proximity to the identified density should be mutated and investigated by electrophysiological recordings.

Yes, the density in the selectivity filter likely represents the Mg²⁺ ion (see response to the previous point), and we have fixed the corresponding labels in all our figures. As suggested, we also attempted to replicate the data from Jean et al. (2002). We have introduced four mutations in D542 of hTRPV6 and found that all generated channel variants were inactive (see Figure R4 below). We noted that this outcome is entirely consistent with other studies using mouse TRPV6 (D541A) and human TRPV6 (D542A)^{8,13-16}. Hence, the idea that D542 contributes to the Mg²⁺ is well consistent with structural data, but this model awaits further functional verification.

Figure R4. Whole-cell currents of WT and mutant hTRPV6 variants expressed in HEK293 cells. Average I-V relationships of the indicated hTRPV6 variants were obtained in the absence of intracellular Mg²⁺, analogous to experiments in Figure 1a,b.

It remains unclear how figure 1d would be explainable by the proposed mechanism? The proposed Mg²⁺ binding site is only formed in the closed state. In the presence of Mg²⁺, Figure 1d shows increasing inward current which slowly grows until ~100s. At this point, ~80% of all

channels are open (by comparing to the current at 0 Mg²⁺). In the open geometry, Mg²⁺ binding should not occur, as the binding site was destroyed by the rotation of the S6. Once the voltage is shifted to +40 mV, all channels close immediately. One would expect a much slower (possibly slower than the opening rate) channel closing rate, as the S6 has first to rotate to form the proposed Mg²⁺ binding site, and only then could Mg²⁺ bind to the proposed site.

After we replaced labels of Ca²⁺ to Mg²⁺ at the D542 site in our figures (see responses to the previous two comments), it has become clear that Mg²⁺ acts according to two mechanisms, channel block through D542 site and allosteric inhibition through the intracellular site.

We believe that a general misunderstanding has arisen regarding our electrophysiological approach. Please note that the release from the block in Figure 1d occurs not in hundreds of seconds but in 0.1 seconds. The results shown in Figure 1d are entirely consistent with previous studies from other laboratories that aimed to demonstrate the effect of Mg²⁺ on whole-cell currents (not the open probability of single channels), reflecting the combined response of thousands of channels. Therefore, the kinetics of Mg²⁺-dependent relaxation of inward TRPV6 currents in Figure 1d cannot be directly linked to the K_d for Mg²⁺ binding, the time required for fine rearrangements in the S6 helix, or channel gate closing. In this context, we do not think that the data in Figure 1d contradicts the structural and MD results.

From the perspective of a physiological role, the competition with Ca²⁺ is missing. I would be very helpful to carry out experiments with increasing extracellular Ca²⁺ concentrations and quantify currents of WT and mutants (including a least one mutant in the selectivity filter).

As shown in Figure R4, four mutations in the selectivity filter (D542) of hTRPV6 created inactive channel variants. Therefore, we are unsure whether the effect of the permeation block of sodium currents by extracellular Ca²⁺ can be clearly separated from its competition with intracellular Mg²⁺ at the internal site, as well as from calmodulin-dependent inactivation of TRPV6 currents.

Figure 5. The D580 mutants are designed to affect Mg²⁺ binding by removing one of the key aspartate residues interacting with Mg²⁺. It is puzzling, how an inward rectification by a Mg²⁺ block at +40 is still possible in these mutants, as it should not bind Mg²⁺.

Please note that we did not claim that inward rectification of TRPV6 was exclusively underpinned by intracellular Mg²⁺. In line with other studies, we observed that WT TRPV6 currents retain rectification in the absence of Mg²⁺ (see Figure 1), indicating that other factors may play a role.

Figure 5e shows a much faster opening kinetics for D580R as compared to all other tested D580 variants, also compared to WT. Something important must be different. Could the authors investigate the reason, or at least speculate on the reason of this observation?

In contrast to other mutant variants examined, the D580R mutation introduces a positive charge, and this could have a more substantial impact on the local distribution of cations, including Mg²⁺.

Figure 6. The D580R mutant exchanges the negatively charged aspartate with the positively charged arginine. In the closed channel state, the arginine side chain will be rotated towards the position, where the Mg²⁺ could be coordinating D498 and D580R. How is it explainable that a Mg²⁺ (at 10 mM) bound to D498 at the putative bind site could stabilize the closed conformation instead of further destabilizing it? In the presence of Mg²⁺, the positive charge at D580R would

be expected to be pushed away (from close to open) enhancing the open state of the channel, possibly also at positive voltages. This seems a contradiction.

As noted on p. 8 lines 203-213, we believe that non-physiologically high concentrations of Mg^{2+} (10 mM) can influence TRPV6 through alternative mechanisms, such as indirectly interacting with phospholipids or other phosphor-containing metabolites like ATP, or directly via low-affinity sites within the TRPV6 protein.

Free Energy: acetate reference system: The authors write on line 201 to 205: " The resulting FES and ΔG^0 values of cation-acetate binding, -3.2 ± 0.5 kJ/mol for Mg^{2+} and $+0.5 \pm 0.3$ kJ/mol for Ca^{2+} , were in reasonable agreement with the experimental data ranging from -7.3 to -3.4 kJ/mol for Mg^{2+} and -6.8 to -2.5 kJ/mol for Ca^{2+} ions, and with the previously reported simulations results (-7.0 to -5.2 kJ/mol for Mg^{2+} and -5.0 to -0.8 kJ/mol for Ca^{2+}). The authors thus report for the acetate binding simulations a value $+0.5$ kJ/mol for Ca^{2+} , which would mean repulsive instead of attraction. Previous computational data reported -5.0 to -0.8 kJ/mol, while experimental data showed -6.8 to -2.5 kJ/mol. This is far from "a reasonable agreement", indicating that the computational approach must suffer from a fundamental shortcoming, suggesting that the data cannot be seen as reliable.

We agree with the Reviewer's comment that the provided data requires clarification, additional explanation and correction. As mentioned in the **Methods** section, accurate modeling of the interactions between divalent cations and carboxylates remains challenging due to the systematic overestimation of binding energies in standard force fields. To overcome this limitation, we adopted the electronic continuum correction (ECC) approach, following the framework established by¹⁷ using a charge scaling factor $SF = 0.8$. Indeed, Ca^{2+} -acetate binding looks underestimated with such parameterization, which was also reported in the original study¹⁷ (ΔG^0 (Ca^{2+} _calculated) = -0.8 kJ/mol, compared to ΔG^0 (Ca^{2+} _experimental) = -3.2 kJ/mol). Notably, in the same study for Mg^{2+} -acetate binding, the ECC approach showed opposite discrepancy to the experiment (ΔG^0 (Mg^{2+} _calculated) = -5.3 kJ/mol, compared to ΔG^0 (Mg^{2+} _experimental) = -3.4 kJ/mol).

To better match the experimental values, we increased the scaling factor for Ca^{2+} to $SF = 0.85$ in the calculations included in the revised manuscript. We also noted that the experimental results were obtained at zero ionic strength, which we had not previously taken into account. Therefore, we added a ΔG_i term to our ΔG^0 estimations for the correction to the same ionic strength (see **equation 1.4** and text above in the **Methods** section). After these modifications, we have verified the correctness of WTMD simulations on the cation-acetate systems and obtained good consistency with the experimental ranges. Magnesium-acetate $\Delta G^0 = -3.7 \pm 0.1$ kJ/mol corresponds reasonably well to the experimental values ranging from -7.3 to -3.4 kJ/mol; calcium-acetate $\Delta G^0 = -3.0 \pm 0.2$ kJ/mol is fitted to the experimental range from -6.8 to -2.5 kJ/mol. For details, see the section "**Validation of simulation protocols**" in the revised **Methods** and **Table S1**. Therefore, in the revised manuscript, we primarily used $SF=0.8$ for Mg^{2+} and $SF=0.85$ for Ca^{2+} , as this combination provides the most physically realistic and experimentally consistent description of the behavior of both cations.

On lines 226 to 230, the authors write: "We observed a significant difference between ΔG^0 for binding of Mg^{2+} (-11 ± 1 kJ/mol) and Ca^{2+} (-3.0 ± 0.2 kJ/mol) to the D489-D580 site, which corresponds to about 20 times higher preference in binding of Mg^{2+} compared to Ca^{2+} . Thus, the D489-D580 site appears to be a magnesium-binding site with high selectivity to Mg^{2+} compared to Ca^{2+} ". The discrepancies observed for the free energy in the acetate example questions the validity of this observation and needs further verification. Also, the stated difference is contradicted by the data shown in Figure 7 and S6, which show no significant differences beyond higher energy barriers for Mg^{2+} , opposing binding of Mg^{2+} . What is the rationale for this statement?

We appreciate the reviewer's thorough examination of our binding energy data and the objective feedback on the comparison with the acetate system. To provide comprehensive answers, we modified the protocols and parameterization of some calculations and additionally performed more than 35 μs of WTMD simulations. We address these concerns below.

1. Although the acetate system (AS) had limitations when calculating Ca^{2+} binding energy with the previously used parameter of $SF=0.8$, our optimized parameter of $SF=0.85$ and zero ionic strength correction helped to achieve agreement with the experimental range. Importantly, this adjustment did not significantly alter the Mg^{2+}/Ca^{2+} selectivity observed in the D489-D580 site of TRPV6_{Mg}. However, the modified calculation protocols make some changes in the numerical values.
2. Additionally, we performed WTMD simulations for the econazole-inhibited hTRPV6 structure (TRPV6_{Eco}, PDB ID: 7S8C)¹⁸, where the D489-D580 site is presented under magnesium-free conditions. The ΔG^0 were found to be -9.4 ± 0.5 kJ/mol for Mg^{2+} and -4.8 ± 0.4 kJ/mol for Ca^{2+} , which corresponds to only 6-fold selectivity. Thus, the geometry of the site notably affects the relative affinity: the optimal D489-D580 arrangement in TRPV6_{Mg} stabilizes a Mg^{2+} rigid octahedral hydration shell, whereas site expansion disrupts this geometry. In the case of Ca, no such change in affinity occurs due to its larger and more flexible coordination sphere. This further confirms that the strong selectivity obtained is not a modeling artifact but rather the consequence of the D489-D580 site adaptation to Mg^{2+} binding in the TRPV6_{Mg} Cryo-EM structure.

To clarify this point, we added the following text to the **Results section**:

The standard binding free energies revealed a striking difference between Mg^{2+} ($\Delta G^0 = -12.9 \pm 0.4$ kJ/mol) and Ca^{2+} ($\Delta G^0 = -3.7 \pm 0.6$ kJ/mol) at the D489-D580 site. This corresponds to an ~35-fold stronger binding preference for Mg^{2+} over Ca^{2+} , demonstrating the D489-D580 site's high selectivity for magnesium. To evaluate the structural basis of the obtained selectivity, we performed the same WTMD simulations for the econazole-inhibited hTRPV6 structure (TRPV6_{Eco}, PDB ID: 7S8C)¹⁹, where the D489-D580 site is presented in the magnesium-free conditions, and the distance between D489 and D580 α atoms is 0.1 nm larger than that in TRPV6_{Mg} (Fig S7a). ΔG^0 were found to be -9.4 ± 0.5 kJ/mol for Mg^{2+} and -4.8 ± 0.4 kJ/mol for Ca^{2+} (Fig S7 and Table S1). Strikingly, this tiny expansion of the site in TRPV6_{Eco} significantly weakened Mg^{2+} binding affinity ($\Delta\Delta G^0 = +3.5 \pm 0.6$ kJ/mol vs. TRPV6_{Mg}), while Ca^{2+} binding remained closely unaffected ($\Delta\Delta G^0 = -1.1 \pm 0.7$ kJ/mol). This differential effect arises from distinct ion coordination chemistry: in the magnesium-optimized geometry of TRPV6_{Mg}, the arrangement of D489 and D580 oxygen atoms stabilizes a near-perfect

octahedral first hydration shell around Mg^{2+} , whereas site expansion disrupts this configuration, reducing Mg^{2+} affinity. Conversely, the larger and more flexible coordination sphere of Ca^{2+} exhibits lower sensitivity to such geometric perturbation. These results demonstrate the crucial role of precise site geometry in Mg^{2+} selectivity and further suggest that in TRPV6_{Mg} it is adopted by the bonded Mg^{2+} .

3. In the revised manuscript, we rerun the WTMD trajectories in a “multiple walkers” configuration and recalculated FES using a reweighting procedure. The new results reveal that Mg^{2+} , compared to Ca^{2+} , exhibits a ~10 kJ/mol higher energy barrier between the water-mediated (State 2) and direct-binding (State 3) local minima (**Fig. 7cd**). In both **Fig. 7c,d** and **Fig. S6h,i**, different projections of the FES show that the free energy minimum for Mg^{2+} is about 10 kJ deeper than for Ca^{2+} , which roughly corresponds to $\Delta\Delta G^0 = 9.2$ kJ/mol, which follows from the reported binding free energy for Mg^{2+} ($\Delta G^0 = -12.9 \pm 0.4$ kJ/mol) and Ca^{2+} ($\Delta G^0 = -3.7 \pm 0.6$ kJ/mol).
4. The robustness of our calculation protocols is further substantiated by the reproducibility across independent simulations of the MG-PS and CA-PS systems. In sum, we obtained consistent results from: (1) the “multiwalker” trajectories reported in the revised manuscript, and (2) two additional single-walker trajectories initiated from bonded and unbonded starting configurations (**Fig. R5**). This consistency across different simulation protocols provides strong validation of our results and confirms that the observed magnesium selectivity is not an artifact of any particular computational approach.

Figure R5. Binding free energy of Mg^{2+} (a) and Ca^{2+} (b) to the D489-D580 site of TRPV6_{Mg} over total simulation time. ΔG^0 values were derived from a reweighting analysis of independent trajectories of the PS system. For the “multiwalker” trajectories (red, with SF=0.8 for Mg^{2+} and Ca^{2+} , yellow for SF=0.85 for Ca^{2+}) total simulation time represents the sum of all 4 walkers. The “single walker” trajectories were initiated from bonded (b, blue, SF=0.8) and unbonded (u, green, SF=0.8) starting configurations. ΔG^0 values were calculated using block analysis applied to the trajectory data from 2 μs to the point on the time axis. Large-scale fluctuations at the beginning are related to poor statistics, while at the end of the trajectories ΔG^0 converged to almost the same values.

Free Energy, PS system: The interpretation of the data could be complicated, if e.g. the sidechain of D489 or D580 would rotate while the divalent ion position remains unchanged. To be sure that this is not the case and phase space remains connected, please report on ch1 of the two aspartate residues.

Indeed, complete sampling of the configuration space is crucial for sustained free energy estimations in the WTMD simulations. It is not only necessary to perfectly sample the biased

CVs, but also any possible CVs that can affect them. The conformations of the D489 and D580 side chains may be such “hidden” CVs. To characterize the available conformational space in the PS system, we performed an unbiased MD simulation without any cation in the site (“MD-free”, 5 μ s). The distributions of the χ_1 (N-C α -C β -C γ) and χ_2 (C α -C β -C γ -O $_{D1}$) dihedral angles for both residues D489 and D580 are plotted in **Fig. S9b,c**. In the MD-free simulation, multiple stable states were observed (gray points). For D489, the angle values were $\chi_1 \approx 50^\circ$ and $\chi_2 \approx -100^\circ/+100^\circ$, while D580 adopted $\chi_1 \approx -60^\circ/180^\circ$ and $\chi_2 \approx -100^\circ/0^\circ/+100^\circ$. To verify that the dihedrals sampling in the WTMD simulations were complete, a reweighting procedure was performed to project FES onto the χ_1 - χ_2 plane. The results for the PS and PS-apo systems are shown in **Fig. S9d-k**. All dihedral angles available in the MD-free case were found to be sampled during WTMD in all trajectories. Additionally, we plotted the χ_1 - χ_2 for the states, when the cations are bonded to the D489-D580 site (red dots at the same panels). Although the statistics is quite poor in this case, it can be seen that in the bonded states WTMD also sampled nearly the same conformational space as in the MD-free trajectory. Thus, we can conclude that the dihedrals were properly sampled in our WTMDs. Also, the independent replicas of WTMD showed very good convergence (**Fig. R5**), that further proves good sampling of “hidden” CVs. We have added the validation performed in the section “**Validation of simulation protocols**” in the **Methods**.

Free Energy method: The divalent ion has been restrained to a cylinder. How was the influence of this restraint corrected for?

The influence of cylindrical restriction on the cation was corrected by including to the ΔG° equation **(1.1)** the standard-state correction term ΔG_V **(1.3)**. Specifically, this correction was incorporated as the ratio of the sampled unbound volume ($l_u S_u$ in **equation 1.3**) to the standard concentration volume V^0 .

Amber was parameterized together with the TIP3P water model. Please cite a study confirming that Amber-99sd-ildn can be combined with the SPC/E water model.

The reviewer is right – Amber99sb-ildn is indeed typically used with the TIP3P water model. At the same time, initial parametrization of this FF was done based on QM calculations in a gas phase^{19,20}, which does not involve any explicit co-parametrization with a particular water model. However, the TIP3P model was further used to validate the FF parameters in MD-simulations. As a result, the combination of Amber99sb-ildn and TIP3P is considered the most reliable. Parameters for Mg²⁺ and Ca²⁺ within the framework of ECC formalism were proposed in¹⁷. They reproduced reasonably well the experimental binding affinities between divalent ions and acetate. For these models, the authors used an Amber99-like FF and the SPC/E water model. To maintain consistency with these results, we employed a similar parameterization setup. In our modeling, we assume two possible effects of SPC/E. First, the stability of the protein structure may be affected by the water model. In²¹, the effects of various combinations of protein and water FFs was investigated, and it was shown that, comparing to Amber99sb+TIP3P, the Amber99sb+SPC/E composition has a very similar root-mean-square deviation (RMSD) of the protein structure in MD-simulations. In our simulations, the fragments of the S5 and S6 helices were used as the site D489-D580 model. This truncated construction would not be stable in MD simulations. In the full-size TRPV6, this site is stabilized by the surrounding protein structure and its heterogeneous water-membrane environment. However, a full protein-membrane-water system would be too large for WTMD simulations, that makes our truncation reasonable. So, we imposed the position restraints on the C α atoms to stabilize the S5-S6 bundle in the water

environment of the PS and PS-apo systems. A side effect of such restraints is the stabilization of the protein secondary structure regardless of the FF combination.

Second, the nonbonded (mainly electrostatic) intermolecular interactions may be modified by the altered water “shielding”. This problem in our model is solved using the ECC approach, which is aimed to correct these effects such as water shielding and polarization. To validate our parametrizations, we calculated ΔG^0 of Mg^{2+}/Ca^{2+} binding to acetate, which turned out to be in a good agreement with experimental values (see above).

It is unexpected to see very different values and block size dependent trends for Mg^{2+} and Ca^{2+} . A change of ~ 9 kJ/mol and changing sign as a function of block size is an alarming signal. Setting the preferred block size value would then allow for obtaining the sought results. Please carefully investigate the source of the striking and alarming difference.

The disparities between Mg^{2+} and Ca^{2+} observed in the block analysis in **Fig. S6b**, **Fig. S7d** and especially in **Fig. S8f** arise from differences in their binding kinetics (**Fig. R6**). For Mg^{2+} , the higher energy barrier at 0.4 nm (**Fig. S8e**) results in slower transitions between bound and unbound states. When the block size used is smaller than the bound-unbound transition timescale, the sampling in the blocks becomes incomplete, potentially including only bound or only unbound state, leading to non-converged ΔG^0 values that may appear positive (e.g., Mg-AS on **Fig. S8f**). The ΔG^0 value converges to a constant value only when the block size exceeds this characteristic time. At the same time, we have not observed this behavior for Ca^{2+} due to its smaller bound-unbound energy barrier, which leads to faster kinetics, better sampling for analogous sizes of blocks and much rapid convergence. Also, reproduction of ΔG^0 in the independent WTMD replica confirms the correctness of our analysis (**Fig. R6**).

Figure R6. Time evolution of the distance between (a) Mg^{2+} and (b) Ca^{2+} and the acetate carboxyl carbon during 1-2 μs of WTMD simulations. Black horizontal lines denote the equilibrium distances for each cation-acetate complex (PMF profiles global minima correspond to 0.31 nm for Mg^{2+} and 0.33 nm for Ca^{2+} , see **Fig. S8e**). Note that Ca^{2+} exhibits much faster bound-unbound kinetics compared to Mg^{2+} .

In the QM calculate, only two water layer were included. It is well known that hydrogen bond mediated water structures orients water molecules way beyond two water layers. If waters are removed beyond two layers, strong perturbances have to expected. Please correct for the is shortcoming or remove the QM part

We are grateful to the reviewer for drawing attention to this point, especially since when describing the systems that were calculated using quantum chemistry methods, an annoying

typo was made in the text of the article: instead of the indicated 4 and 6 water molecules in clusters surrounding Mg^{2+} and Ca^{2+} ions, it was necessary to specify 17 and 20 molecules. water, respectively. Calculations were carried out specifically for such systems. (Since at least two hydration shells of cations were considered, the numbers 4 and 6 are given by mistake – they can only correspond to the first hydration layer.) We apologize for this inaccuracy. This is corrected now in **Methods**.

As for the essence of the question. We fully agree that water molecules closest to cations and carboxyl groups play an important role in determining the electronic-conformational properties of these systems, since water participates in the formation of H-bonds with oxygen atoms of COO- groups and coordinates cations. At the same time, the purpose of these calculations was to estimate the geometry of complexes of COO- groups with cations, primarily the values of the distance between oxygen atoms and cations. These data were necessary for additional verification of the structure of complexes obtained via MD using a ECC force field. To solve such a problem, taking into account (at least) two hydration shells of cations seems justified. According to the literature, 10-14 water molecules surrounded Mg^{2+} and Ca^{2+} ions are already sufficient to ensure the correct geometry of the complexes and, moreover, the frequency of vibration of the bonds in RCOO-molecules coordinating these cations agrees with the observed in experiments^{22,23}.

It should also be noted that the computational protocol that we used to analyze systems simulating the binding site of Mg^{2+} and Ca^{2+} in the TRPV6 channel (two acetate ions/cation) was first tested in similar calculations of simpler and well-studied systems – complexes of both cations with one acetate ion molecule placed in a cluster of water molecules (31 and 22 water molecules, respectively). A comparison of the results obtained in these calculations with the literature data showed an excellent coincidence of the geometry of the complexes under consideration (bond lengths of the cation-oxygen atoms of the acetate ion, coordination numbers, binding mode, etc.). We did not cite these results in the article, since similar ones have already been described in the literature.

Finally, the results of quantum chemical calculations proved useful to us in estimating the degree of polarizability of cations in both Mg^{2+} and Ca^{2+} systems in order to explain the underestimation of ΔG values of formation of acetate ion complexes with Ca^{2+} by the MD method with a scaled force field. A comparison of the electron density distribution on cations suggests that in the case of Ca^{2+} it is necessary to use a higher value of the scaling coefficient in the force field than for magnesium.

It might be interesting to assess, if the Mg^{2+} dependency of channel permeations is conserved across the TRPV family in the same way as its D498 - D580 residue pair.

Yes, D498 and D580 residues are conserved in TRPV channels (see multiple sequence alignment below). Please note that this topic was discussed on p. 12 in the Discussion section: *To our knowledge, the identified regulatory site of TRPV6 has not been discovered in TRP channels before^{24,25}. It is worth noting that several other TRPV channels, including TRPV1, TRPV2, TRPV3, and TRPV5, are known to be modulated by intracellular Mg^{2+} ions²⁶⁻²⁹. Mechanistically, however, such regulatory effects of Mg^{2+} on TRPV channels remain poorly understood. To this end, the present study offers the first structural insight into the role of Mg^{2+} in the inhibition of TRPV6 currents, and future studies are necessary to define whether a similar mechanism is conserved among TRPV channels.*

	Linker S4-S5					S5				S6								
	570	571	572	573	574	575	576	577	578	679	680	681	682	683	684	685	686	687
rTRPV1	E	K	M	I	L	R	D	L	C	I	A	L	M	G	E	T	V	N
	530	531	532	533	534	535	536	537	538	642	643	644	645	646	647	648	649	650
rTRPV2	Q	K	V	I	L	R	D	L	L	I	A	L	M	S	E	T	V	N
	581	582	583	584	585	586	587	588	589	674	675	676	677	678	679	680	681	682
mTRPV3	Q	K	V	I	L	H	D	V	L	I	A	L	M	G	E	T	V	E
	607	608	609	610	611	612	613	614	615	715	716	717	718	719	720	721	722	723
hTRPV4	Q	K	I	L	F	K	D	L	F	I	A	L	M	G	E	T	V	G
	483	484	485	486	487	488	489	490	491	575	576	577	578	579	580	581	582	583
rbTRPV5	Q	K	M	I	F	G	D	L	M	I	A	M	M	G	D	T	H	W
	483	484	485	486	487	488	489	490	491	575	576	577	578	579	580	581	582	583
hTRPV6	Q	K	M	I	F	G	D	L	M	I	A	M	M	G	D	T	HSD	W

Please consistently use nm or Å throughout the manuscript instead of continuously switching units.

We thank Reviewer 5 for pointing out this discrepancy. We have carefully revised the manuscript, now reporting all distance values in nanometers (nm) for consistency.

References

- Hu, H., Bandell, M., Petrus, M. J., Zhu, M. X. & Patapoutian, A. Zinc activates damage-sensing TRPA1 ion channels. *Nat Chem Biol* **5**, 183-190, doi:10.1038/nchembio.146 (2009).
- Radford, R. J. & Lippard, S. J. Chelators for investigating zinc metalloneurochemistry. *Curr Opin Chem Biol* **17**, 129-136, doi:10.1016/j.cbpa.2013.01.009 (2013).
- Bers, D. M., Patton, C. W. & Nuccitelli, R. A practical guide to the preparation of Ca(2+) buffers. *Methods Cell Biol* **99**, 1-26, doi:10.1016/B978-0-12-374841-6.00001-3 (2010).
- John, S., Kim, B., Olcese, R., Goldhaber, J. I. & Ottolia, M. Molecular determinants of pH regulation in the cardiac Na(+)-Ca(2+) exchanger. *J Gen Physiol* **150**, 245-257, doi:10.1085/jgp.201611693 (2018).
- Dweck, D., Reyes-Alfonso, A., Jr. & Potter, J. D. Expanding the range of free calcium regulation in biological solutions. *Anal Biochem* **347**, 303-315, doi:10.1016/j.ab.2005.09.025 (2005).
- Nilius, B. *et al.* Modulation of the epithelial calcium channel, ECaC, by intracellular Ca²⁺. *Cell Calcium* **29**, 417-428, doi:10.1054/ceca.2001.0201 (2001).
- Hoenderop, J. G. *et al.* Function and expression of the epithelial Ca(2+) channel family: comparison of mammalian ECaC1 and 2. *J Physiol* **537**, 747-761 (2001).
- Bodding, M. Voltage-dependent changes of TRPV6-mediated Ca²⁺ currents. *J Biol Chem* **280**, 7022-7029, doi:10.1074/jbc.M410184200 (2005).
- Boone, A. N., Senatore, A., Chemin, J., Monteil, A. & Spafford, J. D. Gd³⁺ and calcium sensitive, sodium leak currents are features of weak membrane-glass seals in patch clamp recordings. *PLoS One* **9**, e98808, doi:10.1371/journal.pone.0098808 (2014).
- Sakipov, S., Sobolevsky, A. I. & Kurnikova, M. G. Ion Permeation Mechanism in Epithelial Calcium Channel TRPV6. *Sci Rep* **8**, 5715, doi:10.1038/s41598-018-23972-5 (2018).

- 11 Saotome, K., Singh, A. K., Yelshanskaya, M. V. & Sobolevsky, A. I. Crystal structure of the epithelial calcium channel TRPV6. *Nature* **534**, 506-511, doi:10.1038/nature17975 nature17975 [pii] (2016).
- 12 Singh, A. K., Saotome, K. & Sobolevsky, A. I. Swapping of transmembrane domains in the epithelial calcium channel TRPV6. *Sci Rep* **7**, 10669, doi:10.1038/s41598-017-10993-9 (2017).
- 13 Bodding, M. & Flockerzi, V. Ca²⁺ dependence of the Ca²⁺-selective TRPV6 channel. *J Biol Chem* **279**, 36546-36552, doi:10.1074/jbc.M404679200 (2004).
- 14 Erler, I., Hirnet, D., Wissenbach, U., Flockerzi, V. & Niemeyer, B. A. Ca²⁺-selective transient receptor potential V channel architecture and function require a specific ankyrin repeat. *J Biol Chem* **279**, 34456-34463, doi:10.1074/jbc.M404778200 (2004).
- 15 Park, E. J. *et al.* Altered biochemical properties of transient receptor potential vanilloid 6 calcium channel by peptide tags. *Biol Pharm Bull* **32**, 1790-1794, doi:10.1248/bpb.32.1790 (2009).
- 16 Weissgerber, P. *et al.* Male fertility depends on Ca²⁺ absorption by TRPV6 in epididymal epithelia. *Sci Signal* **4**, ra27, doi:10.1126/scisignal.2001791 4/171/ra27 [pii] (2011).
- 17 Mendes de Oliveira, D. *et al.* Binding of divalent cations to acetate: molecular simulations guided by Raman spectroscopy. *Phys Chem Chem Phys* **22**, 24014-24027, doi:10.1039/d0cp02987d (2020).
- 18 Neuberger, A., Nadezhdin, K. D. & Sobolevsky, A. I. Structural mechanisms of TRPV6 inhibition by ruthenium red and econazole. *Nat Commun* **12**, 6284, doi:10.1038/s41467-021-26608-x (2021).
- 19 Lindorff-Larsen, K. *et al.* Improved side-chain torsion potentials for the Amber ff99SB protein force field. *Proteins* **78**, 1950-1958, doi:10.1002/prot.22711 (2010).
- 20 Hornak, V. *et al.* Comparison of multiple Amber force fields and development of improved protein backbone parameters. *Proteins* **65**, 712-725, doi:10.1002/prot.21123 (2006).
- 21 Nguyen, T. T., Viet, M. H. & Li, M. S. Effects of water models on binding affinity: evidence from all-atom simulation of binding of tamiflu to A/H5N1 neuraminidase. *ScientificWorldJournal* **2014**, 536084, doi:10.1155/2014/536084 (2014).
- 22 Denton, J. K. *et al.* Molecular-level origin of the carboxylate head group response to divalent metal ion complexation at the air-water interface. *Proc Natl Acad Sci U S A* **116**, 14874-14880, doi:10.1073/pnas.1818600116 (2019).
- 23 Takayanagi, H. *et al.* Stepwise hydration of [CH(3)COOMg](+) studied by cold ion trap infrared spectroscopy: insights into interactions in the magnesium channel selection filters. *Phys Chem Chem Phys* **25**, 23923-23928, doi:10.1039/d3cp00992k (2023).
- 24 Yelshanskaya, M. V. & Sobolevsky, A. I. Ligand-Binding Sites in Vanilloid-Subtype TRP Channels. *Front Pharmacol* **13**, 900623, doi:10.3389/fphar.2022.900623 (2022).
- 25 Talyzina, I. A., Nadezhdin, K. D. & Sobolevsky, A. I. Forty sites of TRP channel regulation. *Curr Opin Chem Biol* **84**, 102550, doi:10.1016/j.cbpa.2024.102550 (2025).
- 26 Cao, X., Ma, L., Yang, F., Wang, K. & Zheng, J. Divalent cations potentiate TRPV1 channel by lowering the heat activation threshold. *J Gen Physiol* **143**, 75-90, doi:10.1085/jgp.201311025 (2014).
- 27 Luo, J., Stewart, R., Berdeaux, R. & Hu, H. Tonic inhibition of TRPV3 by Mg²⁺ in mouse epidermal keratinocytes. *J Invest Dermatol* **132**, 2158-2165, doi:10.1038/jid.2012.144 (2012).
- 28 Mo, X. *et al.* Tyrosine phosphorylation tunes chemical and thermal sensitivity of TRPV2 ion channel. *Elife* **11**, doi:10.7554/eLife.78301 (2022).
- 29 Lee, J., Cha, S. K., Sun, T. J. & Huang, C. L. PIP₂ activates TRPV5 and releases its inhibition by intracellular Mg²⁺. *J Gen Physiol* **126**, 439-451 (2005).

We are very thankful to the Reviewers for their time and effort. We have addressed the final comment of Reviewer #5, with the details outlined below.

Reviewer #1 (Remarks to the Author):

The authors have completed a very thorough response to the critiques of all the reviewers. I do not have any additional concerns.

Reviewer #2 (Remarks to the Author):

After reading the revised version, I think the authors have provided a good response, and all the issues I raised have been properly addressed. I have no further concerns. The revised version is clearly much improved, so I recommend its acceptance.

Reviewer #3 (Remarks to the Author):

The authors have addressed my previous comments thoroughly. They provided additional experimental data, clarified limitations where necessary, and revised the manuscript accordingly. I am satisfied with their responses, and I find the final version of the manuscript suitable for publication.

Reviewer #4 (Remarks to the Author):

Reviewer #5 (Remarks to the Author):

The authors have strongly improved the manuscript, and successfully addressed almost all of my questions. Importantly, the computational data were expanded and clarified to allow for fully comprehending the data.

Minor concerns

Regarding Ca²⁺, please also modify the statement in line 114, replacing calcium with magnesium.

We have modified the statement as suggested.